# OLD BUT GOLD: ADAPTIVE CORESET SELECTION FOR ROBUST DATASET COMPRESSION

## ABSTRACT

The computational and storage costs of large-scale datasets present a significant bottleneck in modern artificial intelligence (AI). While dataset distillation and coreset selection aim to mitigate this by compressing the original datasets into small ones, both have critical limitations. Dataset distillation produces synthetic images that exhibit architectural overfitting and poor transferability to downstream tasks. Conversely, existing coreset selection methods rely on fixed scoring functions, leading to redundant sample selection and performance saturation as the data budget increases. To address these challenges, we propose Adaptive Coreset Selection (ACS), a novel framework that learns an optimal selection strategy for a given budget. ACS employs a multi-stage approach, first building a foundational set of representative samples and then iteratively training models on the selected images to identify hard samples. This adaptive process ensures the final coreset balances representativeness and diversity. We demonstrate the efficacy of ACS on CIFAR-10 and ImageNet, where it outperforms state-of-the-art dataset distillation and coreset selection methods. Notably, on CIFAR-10 with 200 images-per-class, ACS surpasses all baselines by 2%p in validation accuracy and shows superior generalization to downstream tasks, establishing it as a more robust and scalable solution for dataset compression.

## 1 INTRODUCTION

Modern AI systems have demonstrated remarkable success, achieving state-of-the-art performance in various fields, from computer vision (Krizhevsky et al., 2012) to natural language processing (Vaswani et al., 2017). This recent progress largely stems from the growth of computing resources (Kaplan et al., 2020) and the availability of large datasets (Deng et al., 2009). However, this reliance on large-scale data has created a significant bottleneck. The financial and computational costs associated with storing and training on such data present major obstacles (Strubell et al., 2019), limiting the adoption of advanced models. To address this challenge, two primary data compression techniques have emerged: coreset selection (Toneva et al., 2019) and dataset distillation (Wang et al., 2018). Coreset selection chooses representative samples based on scores (*e.g.*, uncertainty scores) from the original dataset to form a smaller subset. More recently, dataset distillation has gained attention as it synthesizes new samples rather than selecting existing ones, offering greater flexibility.

Although dataset distillation has earned its spotlight over coreset selection, synthetic images are specialized for one specific context, limiting their broader applicability. This is because synthetic images are optimized specifically to maximize the validation accuracy of models trained on the distilled datasets. However, optimizing for this single task-specific metric can produce synthetic datasets tailored for one objective, compromising their utility for broader learning tasks such as cross-architecture generalization (Zhao et al., 2021), transfer learning (Yosinski et al., 2014), or domain-generalization (Wang et al., 2022b). In contrast, real images naturally excel at these broader learning tasks without requiring specialized optimization (Yang et al., 2024).

Despite these inherent advantages, existing coreset selection methods fail to show effectiveness across various budgets. While previous literature (Wang et al.) has established that selecting easy samples for small coresets (and hard samples for large coresets) is effective, current approaches employ the same greedy selection strategy regardless of budget size, selecting samples with either the highest or lowest scores. Furthermore, greedy selection leads to redundant selection of similarly

scored samples. As a result, many existing methods perform worse than random selection as the number of images-per-class (IPC) increases, where the diversity becomes more critical relative to the individual quality of each sample. This suggests that effective coreset selection must consider sample importance as a dynamic property that changes based on both the allocated budget and what has already been selected.

To this end, we propose **ACS (Adaptive Coreset Selection)**, a novel framework that learns optimal strategies for selecting informative samples within given budgets. In contrast to methods that use fixed scoring (Toneva et al., 2019; Coleman et al., 2020), ACS adaptively adjusts its selection criteria to construct coresets that consistently outperform both existing coreset selection and state-of-the-art distillation methods. To achieve this, ACS employs a multi-stage strategy that first builds a foundation of representative samples, then iteratively identifies challenging examples based on models trained on the current coreset. This allows ACS to add samples based on the composition of the selected samples, ensuring that new additions complement the current coreset.

We demonstrate the efficacy of ACS on both CIFAR-10 (Krizhevsky et al., 2009) and ImageNet (Deng et al., 2009), benchmarks commonly used to evaluate dataset distillation and coreset selection. Our results demonstrate that ACS outperforms state-of-the-art distillation techniques and existing coreset selection methods on various tasks, particularly at high budgets, establishing it as a more robust and scalable solution. For example, on the CIFAR-10 – 200 IPC setting, ACS surpasses all dataset distillation and coreset selection algorithms in validation accuracy by $2\%$p. We summarize our contributions as follows:

- We reveal limitations in both dataset distillation and existing coreset selection methods, demonstrating that neither approach effectively addresses the challenges of dataset compression across different evaluation scenarios and budget scales (IPC).
- We propose ACS (Adaptive Coreset Selection), a novel framework that dynamically adjusts its selection criteria based on both the allocated budget and the composition of already-selected samples, overcoming the limitations of fixed scoring strategies.
- Extensive experiments show ACS outperforms state-of-the-art methods across CIFAR-10 and ImageNet, achieving up to $2\%$p improvement over the best baselines at 200 IPC while maintaining superiority compared to the random selection baseline.

## 2 RELATED WORK

**Dataset Distillation** aims to synthesize a small dataset such that models trained on it show comparable performance to a model trained on the original dataset. Early works (Zhao et al., 2020; Zhao & Bilen, 2023) operated on small-scale datasets such as CIFAR-10. For example, MTT (Cazenavette et al., 2022), the leading framework for dataset distillation, synthesizes images that mimic the optimization trajectory of real images. However, the immense computational costs makes trajectory matching frameworks infeasible on ImageNet scales. To address this limitation, SRe$^2$L (Yin et al., 2023) proposed utilizing model-inversion to synthesize images. This greatly reduced the computational complexity, and allowed dataset distillation to scale to ImageNet scales. More recently, RDED (Sun et al., 2024) introduced an image-augmentation strategy to synthesize images by augmenting multiple patches of real images into one. However, these methods rely on teacher-generated soft-labels that must be stored along with the images. The storage costs for these labels can be similar to or even larger than storing the images themselves, limiting the practicality of these methods.

**Coreset Selection Methods** identify representative subsets from original datasets without synthetic generation. Early approaches relied on simple sampling strategies (Cochran, 1977) and clustering techniques (Har-Peled & Kushal, 2005; Har-Peled & Mazumdar, 2004; Feldman et al., 2011) to preserve statistical properties. For example, Herding (Welling, 2009) selects samples that have the closest features to the mean of the original dataset. Recent methods analyze training dynamics to score sample importance. The Forgetting score (Toneva et al., 2018) quantifies how frequently a model forgets correct predictions during training, while the EL2N score (Paul et al., 2021) assesses difficulty via the magnitude of the prediction error. These methods rely on fixed heuristic criteria, leading to redundancy as selection budgets increase. This creates a scalability issue, as coreset selection methods suffer from performance degradation, sometimes to a level inferior to the random selection baseline.

**Active learning and curriculum learning** The iterative selection process of ACS can be viewed as a form of self-supervised, pool-based Active Learning (AL). The selection of misclassified samples is analogous to uncertainty sampling, where the model queries points about which it is most uncertain. Unlike traditional AL, which requires a human oracle, the oracle in ACS is the model itself, creating a fully automated loop for identifying informative data. This perspective aligns our work with efforts that frame active learning as a coreset selection problem (Sener & Savarese, 2017), though we propose a novel selection criterion based on learning dynamics rather than geometric properties. Recent works have explored using curriculum learning for dataset distillation. CUDD (Ma et al., 2025) and CCFS (Chen et al., 2025) both employ the curriculum strategy to initialize synthetic images with hard samples. However, both CUDD and CCFS operate in the domain of dataset distillation, and only utilize the selected images as initialization points for synthesis. Furthermore, ACS differs in that it uses an adaptive score that changes based on the already selected images, unlike the fixed scoring used in previous methods.

Table 1: Validation accuracy (%) on CIFAR-10 (50 IPC). The table compares synthetic (MTT, DATM) and real (Random, Forget) data subsets. H and L denote selecting from the high and low scores first, respectively[1], where high score samples change their predictions frequently during training. Real images can show better cross-architecture generalization.

| | | Synth. | | Real | | |
|---|---|---|---|---|---|---|
| Type | Arch | MTT | DATM | Random | Forgetting (H) | Forgetting (L) |
| **Self** | ConvNet | 71.60 | **76.10** | 52.59 | 29.28 | 59.23 |
| **Cross** | ResNet-18 | 60.67 | **61.71** | 53.78 | 28.48 | 59.73 |
| | ResNet-50 | 42.41 | 30.37 | 32.99 | 24.88 | **43.62** |
| | AlexNet | 43.59 | 50.66 | 45.97 | 27.91 | **54.52** |
| | MLP | 34.77 | 34.30 | 36.25 | 23.82 | **41.04** |
| | **Average** | 50.61 | 50.62 | 44.28 | 26.87 | **51.63** |

# 3 PROBLEM FORMULATION

In this section, we first explore the practical benefits of selecting subsets of real images, known as coreset selection (CS), compared to synthetic images generated via dataset distillation (DD). We then examine a key limitation in existing CS methods, which is their difficulty to maintain effectiveness across budgets, especially for high images-per-class (IPC) regions.

## 3.1 PRELIMINARY

Given a dataset $\mathcal{D} = \{(x_i, y_i)\}_{i=1}^N$ with $N$ samples, our goal is to select a coreset $\mathcal{S} \subset \mathcal{D}$ of size $B$ (budget) that maximizes the performance of models trained on $\mathcal{S}$. Let $f_\theta : \mathcal{X} \to \mathcal{Y}$ denote a neural network with parameters $\theta$. The optimal coreset selection can be formulated as:

$$\mathcal{S}^* = \arg \max_{\substack{\mathcal{S} \subset \mathcal{D} \\ |\mathcal{S}|=B}} F(\mathcal{S}), \tag{1}$$

where $F : 2^N \to \mathbb{R}$ measures the quality of a coreset (*e.g.*, validation accuracy of models trained on $\mathcal{S}$). Existing coreset selection methods approximate the quality of coresets by decomposing $F(\mathcal{S})$ into a sum of individual scores:

$$F(\mathcal{S}) \approx \sum_{(x_i, y_i) \in \mathcal{S}} Score(x_i, y_i), \tag{2}$$

where $Score : \mathcal{X} \times \mathcal{Y} \to \mathbb{R}$ is a fixed scoring function (*e.g.*, gradient norm, uncertainty, or forgetting score) computed on each element of the full dataset $\mathcal{D}$. However, these methods employ the same selection strategy for all budgets, *i.e.*, , selecting from the highest or lowest scores. It is observed in the previous literature (Lee & Chung, 2024) that the rule of thumb that uses easy samples for small budgets (and hard samples for large budgets) is effective. Therefore, to fully evaluate the capability of coreset selection methods, it is crucial to choose an appropriate selection criterion (selecting from the highest or lowest scores) for each setting. As we will see in the next section, this simple modification allows coreset selection to achieve competitive performance against dataset distillation.

---

[1]'Forgetting (H)' refers to samples with high forgetting scores, which are often considered hard-to-learn examples. We point out that the high scores do not always refer to hard samples and vice-versa, but differs between methods. For example, sample with high C-score (Raymond-Saez et al., 2022) is an easy one.

## 3.2 PITFALLS OF DATASET DISTILLATION

Dataset distillation (DD) has gained attention as a promising area of research for dataset compression. While it has shown success, its practical value can be context-dependent. Our experiments reveal several key limitations in DD.

**Architectural Overfitting**   One of the main concerns related to DD is architectural overfitting (Moon et al., 2024). While models trained on distilled datasets perform well when the evaluation architecture matches the synthesis architecture, commonly ConvNet (Cazenavette et al., 2022), their performance degrades when evaluated on unseen architectures. Surprisingly, Table 1 shows that with proper selection strategy (selecting from low scores first), the average performance of real-data subsets (Forgetting) is better than synthetic datasets (DATM), as synthetic images show large degradation on unseen architectures. This difference in behavior suggests that synthetic data contain non-generalizable features that are overfitted to a specific architecture and guide the training process differently than real ones.

**Dissimilar Learning Dynamics**   Beyond overfitting, synthetic images induce different learning dynamics from those of real images. We investigate this via modelsoup (Wortsman et al., 2022), which interpolates the parameters of models trained from the same initialization. As shown in Fig. 1, interpolating a model trained on MTT with a model trained on the full dataset results in catastrophic performance degradation. This contrasts with models trained on real data, which interpolate smoothly and suggest that their learned representations are more aligned with those from the full dataset. This raises questions about the reliability of synthetic datasets as training proxies, as models that behave differently from standard training may produce unexpected failures or behaviors when deployed in practice.

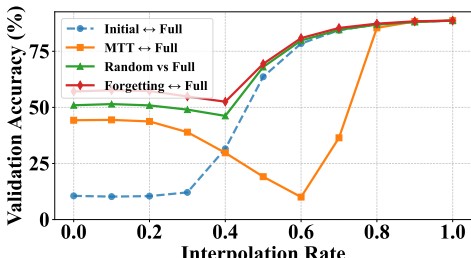

Figure 1: Validation accuracy of models trained on MTT and real images, interpolated with a fully trained model. The model parameters converge to the fully trained model as the interpolation rate reaches 1.0.

To further investigate whether the dissimilar training dynamics result in differences in the functionality of models, we compare various models trained on synthetic versus real images on their domain-generalization capabilities (Wang et al., 2022a). Specifically, we evaluate on CIFAR-10-C (Hendrycks & Dietterich, 2019) and ImageNet-C (Hendrycks & Dietterich, 2019), datasets that share the same classes but with different visual characteristics and corruptions. As shown in Table 2, ResNet-18 trained on synthetic data exhibit worse performance on out-of-distribution (OOD) benchmarks compared to real data. The larger performance degradation on these benchmarks reinforces that synthetic images induce overfit learning dynamics, resulting in models that are less reliable when faced with distribution shifts.

Table 2: Domain generalization performance for CIFAR-10-C and ImageNet-C. All datasets are 50 IPC size. We report mean accuracy (%) of ResNet-18 across corruption types.

| CIFAR-10-C | | ImageNet-C | |
| --- | --- | --- | --- |
| **Method** | **Acc. (%)** | **Method** | **Acc. (%)** |
| MTT | 41.98 | SRe$^2$L | 23.12 |
| DATM | 48.18 | RDED | 26.41 |
| Random | 45.95 | Random | 30.50 |
| **Forgetting** | **50.81** | **Forgetting** | **30.57** |

**Revisiting Coreset Selection**   Our experiments reveal that although dataset distillation excels at generating small datasets capable of training networks from scratch, the synthetic images appear to be specialized for their evaluation context. Specifically, coreset selection shows competitive performance against dataset distillation and better performance on OOD benchmarks. These limitations suggest that for real-world deployment with diverse architectures and potential distribution shifts, carefully selected real images would provide a more robust solution.

## 3.3 THE SCALING PROBLEM OF CORESET SELECTION

While choosing the adequate selection criterion greatly improves performance of coreset selection, it still suffers from the scalability problem. These methods largely outperform random selection at

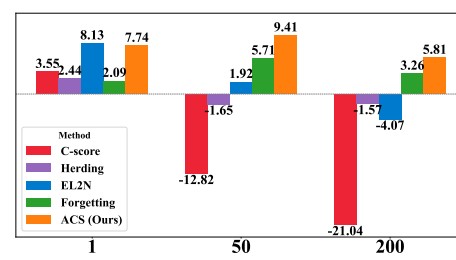 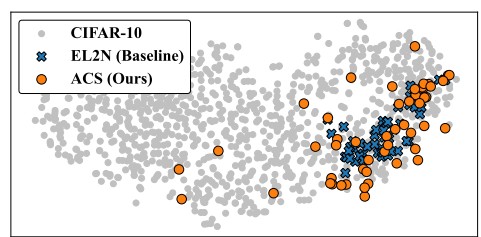

(a) Relative performance of various coreset selection methods on CIFAR-10 against a random baseline across different budget sizes (IPC).

(b) t-SNE (Maaten & Hinton, 2008) visualization of coresets selected by EL2N and ACS.

Figure 2: Empirical evidence of the saturation problem (a) Performance comparison across different budget sizes showing saturation effects, and (b) Feature space visualization revealing tight clustering of EL2N.

low data budgets, but the performance gap quickly saturates as the budget increases. For example, as shown in Fig. 2a, all coreset selection methods surpass random selection at a budget of one images-per-class (IPC). However, at 200 IPC, all but one other than our ACS underperform the random selection baseline, regardless of the selection criterion. This scaling limitation highlights a weakness in existing selection strategies that limit their ability to effectively leverage larger data budgets.

### 3.4 LIMITATIONS OF ELEMENT-WISE GREEDY SELECTION

We argue that the scalability issue stems from the lack of diversity, due to redundant samples being repeatedly selected from fixed scores. Recall from Eq. 2 that existing coreset selection methods approximate the quality of coresets by decomposing $F(\mathcal{S})$ into a sum of individual scores. This decomposition assumes that each sample's contribution to the coreset quality is independent, *i.e.*,

$$F(\mathcal{S}) - F(\mathcal{S} \setminus \{(x_i, y_i)\}) \approx Score(x_i, y_i) \quad \forall \mathcal{S}. \tag{3}$$

In reality, the marginal contribution of a sample depends on what has already been selected. The independence assumption causes greedy methods to select redundant samples, as similar samples have similar scores. Fig. 2b supports our claim. Images selected with EL2N cluster tightly in the feature space, leaving other regions uncovered. In contrast, the coreset selected by ACS contains images in both the regions clustered in EL2N, along with images that are dispersed, representing a diverse selection of representative and challenging samples.

### 3.5 IMPLICATIONS: THE NEED FOR AN ADAPTIVE APPROACH

Our analysis reveals that current coreset selection methods are limited by their selection strategy and element-wise greedy selection. These lead to suboptimal results and saturation of performance at larger budgets, where the importance of diversity increases relative to individual sample quality. This motivates our proposed Adaptive Coreset Selection (ACS) framework, which dynamically adjusts the selection strategy by conditioning its score function on the selected coreset, ensuring representativeness and diversity across all budget sizes.

## 4 ADAPTIVE CORESET SELECTION (ACS)

In this section, we present Adaptive Coreset Selection (ACS), a multi-stage framework that iteratively constructs coresets through adaptive selection. Unlike existing methods that selects all samples at once using a fixed score, ACS considers the coreset holistically and dynamically adjusts its selection strategy based on what has already been selected.

### 4.1 ADAPTIVE DIFFICULTY SCORING

A natural way to make scoring functions context-dependent is conditioning the selected images through model parameterization. We employ the classification loss $\ell(f_\theta(x), y)$ as the scoring metric,

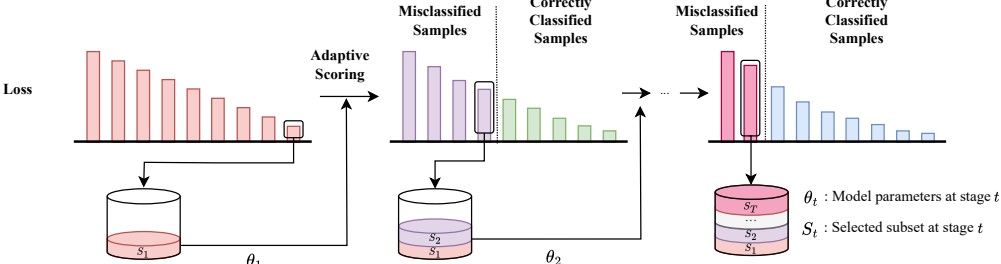

Figure 3: Overview of the Adaptive Coreset Selection (ACS) framework. ACS initializes a foundational coreset $S_1$ with low classification loss samples, then trains a model on $S_1$. At each subsequent iteration, the current model performs adaptive scoring to identify misclassified samples from the remaining data. ACS selects the next subset $S_t$ from these challenging examples and adds it to the coreset. This multi-stage approach iteratively adapts selection criteria to construct diverse and robust coresets.

providing a direct measure of what the model perceives as easy or hard. Then, for any sample $(x, y)$, model $f_\theta$, and cumulative selected images $\mathcal{C}_t$ up to stage $t$, we define the score as

$$Score(x, y; \mathcal{C}_t) = -\ell(f_{\theta_t}(x), y), \tag{4}$$

where $\theta_t$ denotes the parameters obtained by training on the coreset $\mathcal{C}_t$. Low loss values indicate high scoring samples that are easy to classify for the model trained on $\mathcal{C}_t$, while high loss values indicate hard samples where the model struggles. Importantly, the ACS score function explicitly depends on the previously selected images $\mathcal{C}_t$, ensuring that each sample's score is evaluated relative to what has already been selected.

## 4.2 MULTI-STAGE SELECTION STRATEGY

To exploit the adaptive scoring function, we adopt a multi-stage sample selection strategy as illustrated in Fig. 3. We partition the selection into $T$ segments, where each segment selects $\Delta B = B/T$ samples based on the current state of the coreset. In the first segment, we select the easiest samples, defined as those with the highest score under a fully trained model, to establish a foundation of the most representative samples:

$$\mathcal{S}_1 = \underset{\mathcal{S}' \subset \mathcal{D}, |\mathcal{S}'| = \Delta B}{\arg\max} \sum_{(x_i, y_i) \in \mathcal{S}'} Score(x_i, y_i; \emptyset), \tag{5}$$

where $Score(x_i, y_i; \emptyset)$ uses the initial model $\theta_0$ trained on the full dataset.

For subsequent segments $t \in \{2, \ldots, T\}$, we train a model on the accumulated coreset $\mathcal{C}_{t-1} = \cup_{i=1}^{t-1} \mathcal{S}_i$ and identify misclassified samples as follows:

$$\mathcal{M}_t = \left\{ (x_i, y_i) \in \mathcal{D} \setminus \mathcal{C}_{t-1} \mid \arg\max f_{\theta_{t-1}}(x_i) \neq y_i \right\}. \tag{6}$$

From these misclassified samples, we select the subset with the highest aggregate score.

$$\mathcal{S}_t = \underset{\mathcal{S}' \subset \mathcal{M}_t, |\mathcal{S}'| = \Delta B}{\arg\max} \sum_{(x_i, y_i) \in \mathcal{S}'} Score(x_i, y_i; \mathcal{C}_{t-1}). \tag{7}$$

The multi-stage approach naturally prevents the redundancy problem of fixed scoring methods, as similar samples will be correctly classified and excluded from selection. The result is a diverse coreset that captures both the foundational and the challenging samples necessary for model training. We summarize our proposed ACS algorithm in Alg. 1 (see Appendix A).

## 4.3 RATIONALE FOR THE ACS FRAMEWORK

The effectiveness of ACS stems from three key principles:

**1. Easy-to-Hard selection criterion.** ACS automatically incorporates easy samples for low budgets and progressively adds hard samples for higher budgets. This natural progression ensures that

Table 3: Comparison of dataset distillation and coreset selection methods on CIFAR-10 and ImageNet-1K. We report the validation accuracy (%) of ResNet-18 trained on subsets generated by each method. We mark methods that perform worse than random selection in red. The best results are in **bold**, and the second best underlined. Values are presented as mean $\pm$ standard deviation over three runs.

| Category | Method | CIFAR-10 | | | | | Method | ImageNet-1K | | | | |
|---|---|---|---|---|---|---|---|---|---|---|---|---|
| | | 1 IPC | 10 IPC | 50 IPC | 100 IPC | 200 IPC | | 1 IPC | 10 IPC | 50 IPC | 100 IPC | 200 IPC |
| Dataset Distillation | MTT | **35.5**$_{\pm1.2}$ | 48.7$_{\pm0.7}$ | 60.7$_{\pm0.9}$ | - | - | SRe$^2$L | 0.3$_{\pm0.1}$ | 2.3$_{\pm0.7}$ | 5.7$_{\pm0.5}$ | - | - |
| | DATM | 30.6$_{\pm1.0}$ | **51.0**$_{\pm0.5}$ | 61.7$_{\pm0.7}$ | 70.3$_{\pm0.4}$ | 74.3$_{\pm0.4}$ | RDED | 1.8$_{\pm0.4}$ | 12.5$_{\pm1.0}$ | 29.8$_{\pm1.2}$ | 36.3$_{\pm0.6}$ | 38.9$_{\pm0.9}$ |
| Coreset Selection | Random | 16.7$_{\pm1.1}$ | 32.9$_{\pm0.8}$ | 53.8$_{\pm0.6}$ | 63.9$_{\pm1.2}$ | 71.7$_{\pm0.3}$ | Random | 1.0$_{\pm0.2}$ | 6.6$_{\pm1.1}$ | 31.1$_{\pm0.7}$ | 46.2$_{\pm0.5}$ | 57.3$_{\pm0.8}$ |
| | Herding | 20.0$_{\pm0.4}$ | 34.8$_{\pm0.4}$ | 51.0$_{\pm0.8}$ | 58.6$_{\pm0.4}$ | 68.5$_{\pm0.4}$ | Herding | 0.4$_{\pm0.2}$ | 4.0$_{\pm0.8}$ | 19.1$_{\pm0.6}$ | 29.5$_{\pm0.3}$ | 49.8$_{\pm0.7}$ |
| | K-Means | 19.9$_{\pm0.7}$ | 37.5$_{\pm1.1}$ | 53.6$_{\pm0.2}$ | 63.7$_{\pm0.6}$ | 72.2$_{\pm0.5}$ | K-Means | 1.1$_{\pm0.3}$ | 3.7$_{\pm0.4}$ | 15.7$_{\pm0.9}$ | 34.0$_{\pm0.2}$ | 52.8$_{\pm0.5}$ |
| | GraphCut | 18.9$_{\pm1.0}$ | 36.3$_{\pm0.8}$ | 54.6$_{\pm0.3}$ | 64.5$_{\pm0.6}$ | 71.7$_{\pm0.3}$ | GraphCut | 1.8$_{\pm0.1}$ | 12.0$_{\pm0.9}$ | 38.2$_{\pm0.8}$ | 49.1$_{\pm0.6}$ | 57.6$_{\pm0.4}$ |
| | GradMatch | 13.8$_{\pm1.1}$ | 26.8$_{\pm0.6}$ | 37.8$_{\pm0.7}$ | 50.8$_{\pm0.5}$ | 56.4$_{\pm0.8}$ | GradMatch | 0.8$_{\pm0.2}$ | 3.8$_{\pm0.3}$ | 11.3$_{\pm1.0}$ | 19.9$_{\pm0.7}$ | 36.9$_{\pm0.6}$ |
| | Glister | 12.7$_{\pm0.2}$ | 10.0$_{\pm0.0}$ | 10.2$_{\pm0.1}$ | 19.4$_{\pm0.0}$ | 23.5$_{\pm0.1}$ | Glister | 0.5$_{\pm0.2}$ | 3.1$_{\pm0.5}$ | 26.8$_{\pm1.1}$ | 41.2$_{\pm0.7}$ | 39.7$_{\pm1.2}$ |
| | Forgetting | 17.7$_{\pm1.0}$ | 37.5$_{\pm1.3}$ | 59.7$_{\pm0.4}$ | 69.1$_{\pm0.1}$ | 75.5$_{\pm0.3}$ | Forgetting | 1.4$_{\pm0.3}$ | 14.7$_{\pm0.6}$ | 33.3$_{\pm0.9}$ | 45.8$_{\pm0.8}$ | 55.5$_{\pm0.4}$ |
| | EL2N | 23.7$_{\pm0.9}$ | 42.5$_{\pm0.2}$ | 53.4$_{\pm0.0}$ | 62.3$_{\pm0.2}$ | 67.8$_{\pm0.1}$ | EL2N | **2.1**$_{\pm0.5}$ | **17.9**$_{\pm0.7}$ | 40.2$_{\pm0.4}$ | 46.9$_{\pm0.4}$ | 52.7$_{\pm0.9}$ |
| Ours | **ACS** | 24.0$_{\pm0.6}$ | 43.5$_{\pm0.7}$ | **63.2**$_{\pm0.4}$ | **70.8**$_{\pm0.4}$ | **77.5**$_{\pm0.3}$ | **ACS** | 1.5$_{\pm0.3}$ | 15.7$_{\pm0.8}$ | **40.4**$_{\pm0.5}$ | **49.2**$_{\pm0.2}$ | **57.7**$_{\pm0.6}$ |

small coresets focus on the most representative examples, while larger budgets can afford to include challenging samples that refine decision boundaries.

**2. Adaptive scoring.** Unlike existing methods that use fixed scoring (Paul et al., 2021), ACS redefines sample importance based on the already chosen samples. Because the score function is dependent on the previously selected samples $\mathcal{C}_{t-1}$, a sample that appears unimportant initially may become important when evaluated by a model trained on a limited subset.

**3. Natural diversity through multi-stage selection.** By identifying the classified samples and excluding them from selection, ACS can naturally maintain diversity without explicit diversity constraints, even at higher budgets.

## 5 EXPERIMENTS

### 5.1 IMPLEMENTATION DETAILS

**Datasets** Following standard practice in dataset distillation and coreset selection literature, we evaluate all methods on the CIFAR-10 (Krizhevsky et al., 2009) and ImageNet-1K (Deng et al., 2009) to test performance on both small and large-scale scenarios.

**Baselines** We compare ACS against two mainstream methods of dataset compression: dataset distillation and coreset selection. For dataset distillation, we include MTT (Cazenavette et al., 2022), DATM (Guo et al., 2023), SRe$^2$L (Yin et al., 2023), and RDED (Sun et al., 2024). For coreset selection, we compare against Herding (Welling, 2009), K-means clustering (Guo et al., 2022), Gradmatch (Killamsetty et al., 2021a), Glister (Killamsetty et al., 2021b), Forgetting (Toneva et al., 2018), EL2N (Paul et al., 2021), and Graphcut. DATM, RDED, and Graphcut are the state-of-the-art techniques according to the latest benchmarks (Li et al., 2025; Moser et al., 2025) for dataset distillation and coreset selection, respectively. To ensure a fair comparison, all coresets are generated using a ResNet-18 backbone for training from scratch and a ViT-Tiny for transfer learning. For all methods, we adopt the official codebase, including the DeepCore library (Guo et al., 2022), and assume 1% validation set access for Glister.

**Training** All baselines are evaluated using the same training procedure to maintain fairness. We employ ResNet (He et al., 2016) and ViT (Dosovitskiy et al., 2020), two commonly used architectures, for classification tasks. The learning rate is fixed for each IPC setting, except for MTT and DATM, where a separate learning rate is provided. The full details are provided in Appendix B.

**ACS Implementation** We train each intermediate evaluation models with respect to the training details. We start with number of segments $T = 1$ for one IPC, and increase $T$ for higher IPC. Full details are provided in Appendix B.

### 5.2 PERFORMANCE EVALUATION

**Main Results** We compare ACS against state-of-the-art dataset distillation and coreset selection methods on both CIFAR-10 (Krizhevsky et al., 2009) and ImageNet (Deng et al., 2009), using

Table 4: Transfer learning performance on CIFAR-10 and ImageNet-1K by linearly probing a ViT-Base-DINO model. Red denotes performance below the random baseline. The best results are in **bold**, and the second best are underlined. Values are presented as mean $\pm$ standard deviation over three runs.

| Category | Method | CIFAR-10 1 IPC | 5 IPC | 10 IPC | Method | ImageNet-1K 1 IPC | 5 IPC | 10 IPC |
|---|---|---|---|---|---|---|---|---|
| Dataset Distillation | MTT | 11.7±0.4 | 12.4±0.2 | 9.1±0.5 | SRe$^2$L | 25.8±0.4 | 45.5±0.3 | 49.8±0.2 |
| | DATM | 10.9±0.3 | 11.0±0.1 | 11.7±0.2 | RDED | **51.6±0.2** | 58.8±0.4 | 60.2±0.3 |
| Coreset Selection | Random | 35.5±0.3 | 68.2±0.2 | 75.4±0.1 | Random | 35.0±0.3 | 58.3±0.2 | 63.9±0.1 |
| | Herding | 26.8±0.5 | 45.4±0.2 | 43.1±0.5 | Herding | 36.5±0.2 | 58.9±0.3 | 63.7±0.4 |
| | K-Means | 34.0±0.1 | 71.6±0.2 | 77.8±0.1 | K-Means | 35.2±0.4 | 58.7±0.2 | 64.3±0.1 |
| | GraphCut | 50.1±0.2 | 70.3±0.1 | 75.0±0.2 | GraphCut | 36.4±0.3 | 59.4±0.1 | 64.2±0.3 |
| | GradMatch | 32.6±0.5 | 63.6±0.3 | 71.5±0.2 | GradMatch | 36.1±0.5 | 58.9±0.4 | 63.8±0.5 |
| | Glister | 38.1±0.4 | 58.0±0.5 | 67.8±0.3 | Glister | 35.8±0.4 | 58.9±0.2 | 64.0±0.4 |
| | Forgetting | 30.6±0.5 | 62.9±0.4 | 69.4±0.2 | Forgetting | 36.9±0.3 | 58.6±0.2 | 64.0±0.2 |
| | EL2N | 34.5±0.4 | 59.0±0.5 | 65.5±0.3 | EL2N | 36.3±0.2 | 59.0±0.1 | 64.2±0.2 |
| Ours | **ACS** | **52.9±0.2** | **74.4±0.1** | **80.3±0.1** | **ACS** | 37.2±0.2 | **59.7±0.1** | **64.5±0.1** |

ResNet-18 in Table 3. The results show that ACS achieves competitive performance across all IPC settings, and state-of-the-art performance for high budgets (*e.g.*, more than 50 IPC). For example, ACS performs best out of coreset selection methods on CIFAR-10 for 10 IPC and above, performing 2%p better than the next best method (Forgetting) in the 200 IPC setting. Similarly for ImageNet, ACS outperforms other baselines for 50 IPC and above, achieving better performance than GraphCut in the 200 IPC setting. Although baselines show comparable or better performance when the budget is low (1 or 10 IPC), they are not scalable to higher IPC settings. This is because other methods suffer from redundancy, and degrades performance with increasing IPC. In contrast, ACS's adaptive, multi-stage selection facilitates the better selection of samples that complement each other, allowing it to fully utilize the given budget, for both low and high IPC settings.

The results reveal additional interesting observations. First, the effectiveness of dataset distillation versus coreset selection differs between datasets. On CIFAR-10, dataset distillation methods perform well, especially at lower budgets. MTT achieves 35.5% accuracy at 1 IPC, outperforming all coreset selection methods by more than 10%p. However, on ImageNet, these methods fall drastically behind real images, even worse than random selection. This contrast suggests that synthesizing images becomes challenging as domain complexity and resolution increase, making coreset selection more practical for complex datasets. Second, ACS and GraphCut are the only selection methods that consistently outperform the random baseline, while others are inconsistent. This is because ACS and GraphCut are methods that grade sample scores based on other samples. ACS redefines scores for each segment, while GraphCut is a submodular method that minimizes redundancy of selected samples. This again highlights the importance of defining sample importance within the context of other samples, which is crucial to achieving stable performance across various settings.

**Generalization to Downstream Tasks** To test the representativeness of subsets selected by various coreset selection methods, we report the performance of linear-probed ViT-DINO models in Table 4. We observe that models trained on synthetic images show catastrophic performance degradation, implying that they do not represent the distribution of the original dataset. Among coreset selection methods, ACS shows state-of-the-art performance across all settings. Consistent with the finding in Table 3, the performance gain of ACS against random selection is stable with increasing budget. Notably, RDED shows great efficiency at lower budgets for transfer learning. This is because

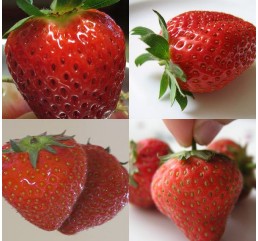 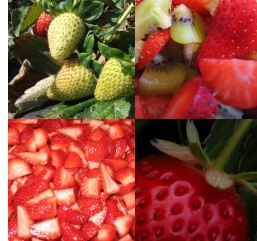

(a) Images from the first segment.  (b) Images from the last segment.

Figure 4: Qualitative results of images selected in the first and last segment using ACS.

RDED aggregates four real images into one, allowing the model to be exposed to a greater diver-

sity of visual features when the number of samples are extremely constrained. Nevertheless, ACS outperforms all methods at the setting of 10 IPC by maintaining representativeness and diversity.

### 5.3 QUALITATIVE ANALYSIS

To understand the characteristics of samples selected by ACS, we visualize images from the first and last segments in Figure 4. The images are selected under the 100 IPC and $T = 3$ setting on ImageNet. The first segment (Figure 4a) contains canonical samples with clear, distinctive features, such as whole strawberries with typical appearance, establishing foundational patterns for the coreset. In contrast, the last segment (Figure 4b) incorporates challenging hard samples that represent edge cases, such as strawberries with atypical appearance or partial occlusion. These samples capture decision boundary complexities that complement the foundational patterns. This demonstrates how ACS maintains diversity through adaptive selection, ensuring a balanced representation of both prototypical and challenging samples in the coreset.

### 5.4 ABLATION STUDIES

**Backbone architecture** The choice of backbone for selecting images plays a critical role in the ACS algorithm. To investigate the effect of backbone choice, we evaluate the performance of ACS when using ConvNet, ResNet-18, and ResNet-34 backbones in Table 5. Using ConvNet, a very small architecture, achieves comparable or superior performance to ResNet-18, while ResNet-34 degrades performance. This indicates smaller models are better suited for our algorithm, as larger models require more

Table 5: Ablation study of ACS on the backbone architecture used for selecting images. Small models result in informative samples compared to using large models.

| IPC | ConvNet | ResNet18 | ResNet34 |
|-----|---------|----------|----------|
| 10  | 42.49   | **43.91** | 40.90   |
| 100 | **71.08** | 70.79   | 66.08   |

data to learn meaningful representations (Kaplan et al., 2020). While ConvNet's competitive performance suggests potential gains from backbone optimization, we adopt ResNet-18 as the standard backbone to ensure fair comparison across coreset selection methods.

**Number of segments** We study the impact of the multi-stage strategy by varying $T$ from one to ten across different budgets in Table 6. We observe that using $T > 1$ always shows greater performance than $T = 1$, validating the effectiveness of our multi-stage strategy. Furthermore, lower values of $T$ perform better at small budgets, while higher $T$ values excel at larger budgets. At 10 IPC, using $T = 5$ outperforms $T = 10$ by 1%p, as excessive segmentation leaves too few samples per segment to train

Table 6: Ablation study of ACS on the number of segments $T$. We report the performance of ResNet-18 trained on CIFAR-10 coresets selected with varying number of segments.

| IPC | T=1 | T=2 | T=5 | T=10 |
|-----|-----|-----|-----|------|
| 10  | 38.99 | 41.85 | **43.48** | 42.44 |
| 100 | 63.31 | 67.38 | 67.90 | **69.48** |

meaningful intermediate models. Conversely, at 100 IPC, $T = 10$ surpasses $T = 5$ by 1.5%p, benefiting from the fine-grained curriculum effects. This trend suggests that while more segments provide better curriculum effects at larger budgets, each segment should still contain sufficient samples to support meaningful misclassification patterns.

## 6 CONCLUSION

In this work, we revealed fundamental limitations in both dataset distillation and existing coreset selection methods. Dataset distillation suffers from architectural overfitting and poor transferability, while coreset selection methods fail to fully utilize their scoring and selection strategies. To address these challenges, we proposed Adaptive Coreset Selection (ACS), a multi-stage framework that dynamically adjusts selection strategies based on coreset composition. Extensive experiments on CIFAR-10 and ImageNet show ACS consistently outperforms baselines across all scales, achieving state-of-the-art performance while maintaining superior generalization to downstream tasks. Our results highlight the importance of sophisticated real data selection, establishing ACS as a robust and practical solution for dataset compression.

## 7 REPRODUCIBILITY STATEMENT

We ensure reproducibility by providing detailed documentation of all hyperparameters, schedulers, and experimental setups in Appendix B. All datasets and architectures used in this paper are publicly available.

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

# APPENDIX

## A ACS ALGORITHM PSEUDOCODE

---

**Algorithm 1: Adaptive Coreset Selection (ACS)**

---

1: **Input:** Original dataset $\mathcal{D}$, total budget $B$, number of stages $T$, initial model parameters $\theta_0$.
2: **Output:** Coreset $\mathcal{S}$ of size $B$.
3: **Notation:** $\Delta B = B/T$ is the stage budget, $\mathcal{M}_t$ is the misclassified set at stage $t$.
4: **Initialize:** $\mathcal{C}_0 \leftarrow \emptyset$
5: Train initial model with parameters $\theta_0$ on full dataset $\mathcal{D}$.
6: Select $\Delta B$ samples from $\mathcal{D}$ with the highest $Score(x, y; \emptyset)$ using $\theta_0$ (Eq. 2). Let this set be $\mathcal{S}_1$.
7: $\mathcal{C}_1 \leftarrow \mathcal{S}_1$
8: **for** $t = 2$ **to** $T$ **do**
9:     Train a new model with parameters $\theta_{t-1}$ on the current coreset $\mathcal{C}_{t-1}$.
10:     Identify misclassified samples $\mathcal{M}_t$ from $\mathcal{D} \setminus \mathcal{C}_{t-1}$ using $\theta_{t-1}$ (Eq. 4).
11:     Select $\Delta B$ samples from $\mathcal{M}_t$ with the highest $Score(x, y; \mathcal{C}_{t-1})$ to form subset $\mathcal{S}_t$ (Eq. 5).
12:     $\mathcal{C}_t \leftarrow \mathcal{C}_{t-1} \cup \mathcal{S}_t$
13: **end for**
14: **return** $S = \mathcal{C}_T$

---

Algorithm 1 presents the complete Adaptive Coreset Selection (ACS) procedure. ACS operates in $T$ stages, selecting $\Delta B = B/T$ samples per stage to construct a final coreset of size $B$.

The algorithm begins by training an initial model on the full dataset and selecting the first subset $\mathcal{S}_1$ comprising samples with the lowest initial training loss (lines 5-7). This foundational subset captures easily learnable patterns that provide stable training initialization.

For subsequent stages $t = 2$ to $T$, ACS follows an iterative refinement process (lines 8-16). At each stage, a model is trained on the current coreset $\mathcal{C}_{t-1}$ and used to identify misclassified samples $\mathcal{M}_t$ from the remaining data. The selection strategy then differs based on task type: for training from scratch, ACS selects samples from $\mathcal{M}_t$ with the lowest context-aware scores (Equation 5), prioritizing challenging examples that complement the existing coreset. For downstream tasks, random selection from misclassified samples suffices due to the robust representations provided by pretrained models.

This curriculum-based approach ensures progressive difficulty increase while maintaining diversity through context-aware scoring, resulting in coresets that effectively capture both foundational and challenging aspects of the original dataset.

# B ACS IMPLEMENTATION

We provide the full hyperparameters needed to implement ACS in Table 7, Table 8, and Table 9.

Table 7: Hyperparameters for ACS experiments on CIFAR-10. Augmentations include Differentiable Siamese Augmentation (DSA) and CutMix (CM).

| Hyperparameter | 1 IPC | 10 IPC | 50 IPC | 100 IPC | 200 IPC |
|---|---|---|---|---|---|
| Learning Rate | 0.025 | 0.015 | 0.03 | 0.025 | 0.05 |
| Optimizer | SGD | SGD | SGD | SGD | SGD |
| Epochs | 1000 | 1000 | 1000 | 1000 | 1000 |
| Batch Size | 64 | 64 | 64 | 64 | 64 |
| Augmentations | DSA, CM | DSA, CM | DSA, CM | DSA, CM | DSA, CM |
| Segments (T) | 1 | 5 | 5 | 8 | 10 |

Table 8: Hyperparameters for ACS on ImageNet-1K, where the augmentation used is Random-sized Recrop (RR).

| Hyperparameter | 1 IPC | 10 IPC | 50 IPC | 100 IPC | 200 IPC |
|---|---|---|---|---|---|
| Learning Rate | 0.001 | 0.001 | 0.001 | 0.001 | 0.001 |
| Optimizer | AdamW | AdamW | AdamW | AdamW | AdamW |
| Epochs | 300 | 300 | 300 | 300 | 300 |
| Batch Size | 128 | 128 | 128 | 128 | 128 |
| Augmentations | RR | RR | RR | RR | RR |
| Segments (T) | 1 | 1 | 2 | 3 | 4 |

Table 9: Hyperparameters for Transfer Learning ACS experiments on CIFAR-10 and ImageNet. The augmentation is Centercrop (CC), and listed parameters are for the linear probing evaluation.

| Hyperparameter | CIFAR-10 | | | ImageNet | | |
| | 1 IPC | 5 IPC | 10 IPC | 1 IPC | 5 IPC | 10 IPC |
|---|---|---|---|---|---|---|
| Optimizer | AdamW | AdamW | AdamW | AdamW | AdamW | AdamW |
| Learning Rate | 0.001 | 0.001 | 0.001 | 0.001 | 0.001 | 0.001 |
| Epochs (Probing) | 20 | 20 | 20 | 20 | 20 | 20 |
| Augmentations | CC | CC | CC | CC | CC | CC |
| Batch Size | 64 | 64 | 64 | 128 | 128 | 128 |
| Segments (T) | 1 | 1 | 2 | 1 | 1 | 1 |

