# OpenReview forum: "Old but Gold: Adaptive Coreset Selection for Robust Dataset Compression"
_ICLR.cc/2026/Conference — Submitted to ICLR 2026_

### Official Review · Reviewer_gVDY · 2025-10-28

**Soundness:** 2
**Presentation:** 3
**Contribution:** 2
**Rating:** 4
**Confidence:** 4

**Summary:**

The paper proposes Adaptive Coreset Selection (ACS). ACS iteratively builds a coreset of examples. First selects representative examples and then iteratively adds hard examples as the model gets trained. ACS is compared against coreset selection and dataset distillation methods. Methods are evaluated on the CIFAR10 and ImageNet.

**Strengths:**

- Paper is well written and understandable.
- The method proposed challenges the idea of static coreset selection with good arguments.
- Authors do a good job in showing weaknesses of related work.

**Weaknesses:**

Insufficient evidence for cost claims and practicality: Paper asserts high “financial and computational costs” but provides no concrete measurements (Questions 1).

Comparison to related work: Paper seems to claim novelty on building the coreset dynamically, however that has been proposed in the past (Questions 2, 3). Additionally, comparison to more recent methods would benefit the paper [2,3,4,6].

Decision reasoning: choices such as starting with easy examples are stated without evidence or reasoning, no experiments as why not to start with hard examples.

Please find further weaknesses in Questions.

**Questions:**

1. “The financial and computational costs associated with storing and training on such data present major obstacles (Strubell et al., 2019), limiting the adoption of advanced models.” (line 036). However, there are no measurements supporting this claim. For example, could you report time-to-accuracy and wall-clock time benchmarks? How are costs reduced if one must first pretrain an entire model on the full dataset? What is the overhead of using ACS? How much can training time be reduced end-to-end?

 2. “Current approaches employ the same greedy selection strategy regardless of budget size” (Line 053): [3] have adapted these methods to perform dynamic sampling during training and to sample from both static and dynamic distributions.


3. Could you clarify how your method differs from Forgetting as adapted in [3]


4. “Fig. 2b supports our claim. Images selected with EL2N cluster tightly in the feature space, leaving other regions uncovered.” (Line 246): If the goal is to cover all regions, why not maximize distribution coverage using an explicit metric (e.g. Maximum Mean Discrepency (MMD))? More broadly, how do you determine that maximizing distribution coverage is desirable, shouldn’t this depend on the data distribution and the subset size?


5. “Unlike existing methods that selects all samples at once using a fixed score, ACS considers the coreset holistically and dynamically adjusts it selection strategy based on what has already been selected.” (Line 263): Dynamic sampling has already been proposed; see [5-6] and [3]. Please clarify the novelty relative to these approaches.


6. “In the first segment, we select the easiest samples,” (Line 298). Why begin with easy examples, and how is this choice validated? [1] provide theoretical conditions under which one should start with easy versus hard examples.


7. What is the motivation for stratified sampling? How do you determine that having equal numbers per class is preferable to allocating more samples to class A than to class B? For example, when in your case having 10 IPC, why is it not optimal to take 15 images of class A and only 5 images of class B?


8. In general, reporting IPC together with the selection/pruning ratio is more informative. For CIFAR-10/100 this is easy to compute, but otherwise stating that a model is trained on “2%” of the data is more informative than stating it is trained on 10 IPC.


9. “This indicates smaller models are better suited for our algorithm, as larger models require more data to learn meaningful representations” (Line 455): I do not believe this follows from Table 5. To support the claim, it would help to show performance when each model is trained on the entire dataset; otherwise, the result could reflect architectural differences rather than an interaction with ACS. Additionally, given scaling laws, shouldn’t one prefer a method that scales with the number of model parameters?


[1] Sorscher, Ben, et al. "Beyond neural scaling laws: beating power law scaling via data pruning." Advances in Neural Information Processing Systems 35 (2022): 19523-19536.

[2] Kolossov, Germain, Andrea Montanari, and Pulkit Tandon. "Towards a statistical theory of data selection under weak supervision." (ICLR’24)

[3] Okanovic, Patrik, et al. "Repeated Random Sampling for Minimizing the Time-to-Accuracy of Learning." The Twelfth International Conference on Learning Representations.

[4] Abbas, Amro Kamal Mohamed, et al. "Effective pruning of web-scale datasets based on complexity of concept clusters." The Twelfth International Conference on Learning Representations.

[5] Mirzasoleiman, Baharan, Jeff Bilmes, and Jure Leskovec. "Coresets for data-efficient training of machine learning models." International Conference on Machine Learning. PMLR, 2020.

[6] Qin, Ziheng, et al. "InfoBatch: Lossless Training Speed Up by Unbiased Dynamic Data Pruning." The Twelfth International Conference on Learning Representations.

---

> ### Author Response · Authors · 2025-11-21
>
> Reviewer NQnR:
> We sincerely thank Reviewer NQnR for their thorough and insightful review. We appreciate the opportunity to clarify the distinction between our work and dynamic training methods, and we value the constructive feedback on our experimental protocols. We address each specific point below.
>
> **[Q1] For example, could you report time-to-accuracy and wall-clock time benchmarks? How are costs reduced if one must first pretrain an entire model on the full dataset? What is the overhead of using ACS? How much can training time be reduced end-to-end?**
>
> We appreciate the reviewer’s concern regarding the practical efficiency of our method. We have provided a detailed wall-clock time analysis in the Common Answer: Computational Cost and Wall-Clock Analysis.
> Summary of Findings: Contrary to the intuition that iterative training creates a bottleneck, our results show that ACS is highly scalable. On ImageNet-1K (100 IPC), ACS is actually faster (6h) than the standard Forgetting baseline (7.7h). This is because ACS trains models on the tiny, growing coreset (e.g., <5% of data) rather than the full dataset (1.2M images). Thus, the "overhead" is minimal compared to the cost of full-dataset training required by other selection metrics.
>
>
> **[Q2 and Q3] Okanovic, Patrik, et al.  have adapted these methods to perform dynamic sampling during training and to sample from both static and dynamic distributions.**
>
> We appreciate the reviewer pointing out the relevant work by Okanovic et al. [1]. We would like to clarify the scope of our claim and the fundamental distinction between our respective problem settings.
> 1. Dataset Compression vs. Training Acceleration: Our statement regarding "greedy selection strategies" refers specifically to the domain of Dataset Compression, where the objective is to identify a single, fixed subset (e.g., 10% of the data) that can be permanently stored and shared to replace the full dataset. In contrast, the method proposed in [1] focuses on Training Acceleration (Time-to-Accuracy) by dynamically sampling different subsets during the training process. In their setting, the model effectively accesses the full data distribution over time, whereas ACS operates under the constraint that the final coreset must be static and portable.
> 2. Deterministic vs. Probabilistic Methodology:
> Okanovic et al. [1] (RS2): Relies on Repeated Random Sampling, where a new random subset is drawn from the full dataset at every epoch to approximate the full distribution.
> Forgetting-RS (Baseline in [1]): As described in [1], this method constructs a probability distribution based on forgetting scores and samples from it repeatedly. This is a probabilistic approach designed to expose the model to the full dataset over time, prioritizing hard examples.
> ACS (Ours): Is deterministically adaptive and constructive. We do not sample probabilistically; we iteratively test the current coreset's coverage and explicitly add samples that are misclassified by the current coreset. This ensures that the final static subset fills specific "blind spots" in the decision boundary—a necessary step when the model will only ever see that specific subset.
> 3. Dependency on Full Dataset Access: The dynamic methods in [1] (both RS2 and Forgetting-RS) require access to the full dataset during training to resample batches. ACS restricts this overhead to the one-time compression phase. Once the ACS coreset is generated, it is a standalone artifact. A model training on an ACS coreset does not need to access the full database, making it a true compression solution rather than a curriculum learning schedule.
>
>
> [1] Okanovic, Patrik, et al. "Repeated Random Sampling for Minimizing the Time-to-Accuracy of Learning." The Twelfth International Conference on Learning Representations.

---

> ### Author Response · Authors · 2025-11-21
>
> **[Q4]  why not maximize distribution coverage using an explicit metric (e.g. Maximum Mean Discrepency (MMD))? More broadly, how do you determine that maximizing distribution coverage is desirable, shouldn’t this depend on the data distribution and the subset size?**
>
> We appreciate this insightful question regarding the definition and optimization of "coverage."
> Please refer to the common answers to see the details of redundancy.
> 1. Geometric Coverage vs. Decision Boundary Coverage While maximizing geometric metrics like Maximum Mean Discrepancy (MMD) ensures feature space coverage, it does not necessarily guarantee training efficiency. Pure distribution matching (e.g., Herding ) focuses on geometric representativeness but ignores the model's learning dynamics.
> Empirical Evidence: We included Herding (which minimizes a form of distribution discrepancy) as a baseline in Table 3. ACS significantly outperforms Herding (e.g., 77.5% vs. 68.5% on CIFAR-10 200 IPC), suggesting that learning-aware coverage (targeting misclassified regions) is superior to static geometric coverage for discriminative tasks. ACS focuses on covering the decision boundary complexities that static metrics miss.
> 2. Desirability of Coverage Depends on Budget We fully agree that maximizing coverage is not always desirable and depends heavily on the budget. This is exactly the motivation behind ACS's adaptive design:
> Low Budget (Focus on Modes): At small budgets, covering outliers is harmful because the model hasn't learned the core concepts. ACS handles this by selecting "easy" (prototypical) samples in the first stage, effectively prioritizing density over broad coverage.
> High Budget (Focus on Edge Cases): As the budget increases, ACS naturally expands "coverage" to the decision boundaries (misclassified samples) only after the core concepts are mastered.
> Conclusion: Unlike explicit MMD maximization which enforces a fixed coverage definition regardless of budget, ACS allows the model's current competence to dictate which regions need to be covered next.
>
> **[Q5]  Dynamic sampling has already been proposed. Please clarify the novelty relative to these approaches.**
>
> We sincerely thank the reviewer for this astute observation. We agree that submodular methods (such as CRAIG) theoretically encode redundancy reduction. We apologize if our initial manuscript implied otherwise; our intention was to highlight the limitations of fixed-score greedy methods (like standard EL2N).
> 1. Geometric vs. Functional Redundancy Submodular methods like CRAIG typically manage redundancy in the Gradient Space (geometric proxy). ACS addresses redundancy in the Functional Space (Decision Boundary). A sample might be geometrically distinct yet "easy" for the model to learn given existing data. By testing against a trained model state, ACS removes "functional redundancy" that geometric submodularity might retain.
> 2. Optimization Strategy: Global vs. Conditional We view ACS as a conditional evolution of the submodular principle. While CRAIG solves a "Cover" problem to match global statistics, ACS solves a "Repair" problem—explicitly hunting for specific "holes" in the decision boundary left by the previous stage.
> 3. Empirical Verification We have added a quantitative redundancy analysis using Vendi Scores in the Common Answer: Quantitative Analysis of Sample Redundancy. The results confirm that ACS achieves significantly lower redundancy (higher Vendi Score) compared to standard baselines, including submodular methods such as GraphCut.
>
>
>
>
>
>
> **[Q6] Why begin with easy examples, and how is this choice validated?**
>
> We appreciate the reviewer referring us to the theoretical insights in Sorscher et al. Our decision to initialize with "easy" samples is deliberate and aligns with their findings.
> 1. Alignment with Sorscher et al.: Sorscher et al. demonstrate that in low-data regimes, retaining "easy" (prototypical) samples is optimal because they best approximate class geometry. In the first segment of ACS, we operate with a tiny fraction of the budget (e.g., 10 images per class). Therefore, prioritizing easy samples to establish stable class prototypes is theoretically consistent with .
> 2. The "Cold Start" Stability Problem” Starting with easy samples is crucial for the stability of our adaptive loop. If we started with Hard Samples (often outliers/noise), the initial model θ1​ would learn a chaotic decision boundary, making the subsequent "misclassification" signal noisy and unreliable. By anchoring the boundary around core modes first, we ensure subsequent "hard" samples are genuinely informative.
>
> **3. Empirical Results**
> We have conducted ablations studies in our common answer. The results show that starting with easy samples is crucial.

---

> ### Author Response · Authors · 2025-11-21
>
> **[Q7 and Q8] What is the motivation for stratified sampling? How do you determine that having equal numbers per class is preferable to allocating more samples to class A than to class B?/ In general, reporting IPC together with the selection/pruning ratio is more informative.**
>
> We thank the reviewer for these insightful questions regarding our sampling distribution and reporting standards.
> 1. Motivation for Stratified Sampling Our decision to use stratified sampling (equal images per class) was driven by the need to align strictly with the established Dataset Distillation evaluation protocol. Major baselines (MTT, RDED) universally benchmark at fixed budgets (1, 10, 50 IPC). To support our central claim—that Coreset Selection (real data) outperforms Dataset Distillation (synthetic data)—it was crucial to operate in the exact same regime.
> 2. Future Work and Reporting Improvements We agree that dynamic budget allocation (e.g., more samples for harder classes) is a compelling direction. Regarding reporting, we appreciate the suggestion to include pruning ratios alongside IPC. We will update our final figures/tables to report both (e.g., "10 IPC / 0.2%") to provide better context for the compression level.
>
>
> **[Q9] I do not believe this follows from Table 5. To support the claim, it would help to show performance when each model is trained on the entire dataset; otherwise, the result could reflect architectural differences rather than an interaction with ACS.**
>
>
> We appreciate the reviewer's insightful question regarding the interaction between model size and ACS selection quality. We agree that the claim "smaller models are better suited" requires more nuance.
>
> **1. Clarification on "Selection" vs. "Training" Models**
> Our observation in Line 455 refers specifically to the *Selection Model* (the proxy model used to calculate loss scores during the search phase), not necessarily the final target model.
> * **Hypothesis:** In the very early stages of ACS (e.g., when the coreset size is extremely small, $<100$ images total), large over-parameterized models (like ResNet-34) may overfit the noise or fail to converge to a stable decision boundary compared to compact models (ConvNet). If the proxy model does not learn a stable boundary, its "misclassification" signal becomes random, degrading the selection quality for the next stage.
> * **Evidence:** Table 5 shows that coresets selected by the smaller ConvNet transfer effectively to larger models, suggesting the "small proxy" strategy is robust.
>
> **2. Scaling Laws and Future Work**
> We agree with the reviewer that ideally, a selection method should scale with model parameters. The apparent degradation with ResNet-34 in our low-budget experiments likely reflects the "data starvation" regime where scaling laws break down because the sample size is insufficient to constrain the model parameters. We acknowledge that for larger budgets (e.g., >20% of the dataset), using larger selection models would likely become optimal. We will refine the text to clarify that this observation is specific to the *extreme low-data regime* characteristic of dataset distillation benchmarks.

---

### Official Review · Reviewer_o3t9 · 2025-10-31

**Soundness:** 3
**Presentation:** 3
**Contribution:** 2
**Rating:** 6
**Confidence:** 3

**Summary:**

This paper tackles the limitations of fixed-score coreset selection and dataset distillation for dataset compression, proposing an Adaptive Coreset Selection (ACS) framework. ACS employs a multi-stage strategy: firstly it selects representative easy samples, then iteratively adds hard examples identified through models which trained on the current subset, progressively constructing a diverse and representative coreset. Comprehensive experiments on CIFAR-10 and ImageNet demonstrate that ACS outperforms state-of-the-art dataset distillation and coreset selection methods, especially at large data budgets, and achieves better generalization on downstream and OOD tasks.

**Strengths:**

1. **Clear identification of limitations:** Both dataset distillation and standard coreset selection methods have clear limits. Dataset distillation often leads to overfitting because it uses synthetic data. Standard coreset methods rely on fixed scoring rules, which can pick too many similar samples. This causes redundancy and makes the selection less effective.
2. **Methodological innovation**: ACS uses a multi-stage, adaptive scoring method which updates how important each sample is as the coreset grows. Most earlier approaches assume samples are independent, but ACS breaks that assumption. The paper explains this idea clearly. It also shows it well in the pipeline diagram.
3. **Comprehensive experimental validation**: The paper provides robust empirical evidence  on both CIFAR-10 and ImageNet for various images-per-class setups, covering accuracy, transferability, OOD robustness, and ablation on architectural backbones and segmentation strategies.
4. **Qualitative analysis**: Visualization of selected images from early and late stages demonstrates that ACS achieves both representative base coverage and challenging sample inclusion—evidence for improved diversity.
5. **Reproducibility**: Hyperparameters are fully provided in Appendix B, and the algorithm is presented in clear pseudocode, supporting claims of transparency in implementation.

**Weaknesses:**

1. **Lack of formal theoretical analysis of generalization**: Apart from the intuitive rationale and empirical results, there’s no formal proof or bound concerning why ACS-selected subsets would achieve better generalization or diversity than other approaches. For instance, there’s no analysis of marginal gains or redundancy reduction, nor is there a clear link to submodular optimization or curriculum learning theory.
2. **Scalability and compute cost**: While the paper claims ACS is scalable, the multi-stage process (involving multiple model trainings) likely incurs noticeably larger compute than single-pass, fixed-score methods. There is no analysis (empirical or theoretical) on computational cost versus performance, nor is wall-clock time or total FLOPs reported in any table.
3. **Algorithmic ablations lacking**: Apart from backbone and segment number (see **Table 5** and **Table 6**), there is no thorough ablation of, for example, whether selecting only from misclassified samples at later stages is essential, or how sensitive ACS is to the exclusion of hard or easy samples in various regimes. This makes it hard to judge whether the method’s advantage is robust to minor tweaks.
4. **Notational clarity**: While the pseudocode in Appendix A and equations throughout are generally sound, some ambiguity remains. For example, the notational switch between $\mathcal{C}_t$ (cumulative coreset) and $\mathcal{S}_t$ (current segment) could be more visually separated throughout the main text and algorithm. Additionally, the loss function notation is at times overloaded.

**Questions:**

1. **Choice and generality of scoring function**: Have you experimented with alternative context-aware scoring metrics beyond classification loss, such as prediction margin, uncertainty (entropy), or uncertainty-gradient products? Would such measures yield better/robuster coresets? Please comment on the generalizability of the principle.
3. **Diversity metrics**: Beyond qualitative t-SNE and segment visualizations, have you quantified the diversity of ACS-selected samples (e.g., feature-space coverage, cluster spread, redundancy metrics)? Could you add such assessments to strengthen the evidence for ACS’s claims?
5. **Hyperparameter scheduling**: Is there any principled scheme to set the number of segments ($T$) based on data or model properties? Have you considered automatic tuning or meta-learning approaches?
6. **Sensitivity to model/backbone choice**: Table 5 suggests varying robustness, but how stable is ACS performance across a wider range of architectures, especially for current large transformer models? Is there a risk of over-specialization?

---

> ### Author Response · Authors · 2025-11-21
>
> Reviwer o3t9:
>
> We sincerely thank Reviewer o3t9 for their positive assessment and constructive feedback. We are encouraged that you found our identification of the "fixed-score limitation" to be a clear and valuable contribution. We also appreciate your recognition of our method’s intuitive grounding in curriculum learning. We address your questions regarding theory, scalability, and ablation details below.
>
> **[W1] Lack of formal theoretical analysis of generalization: Apart from the intuitive rationale and empirical results, there’s no formal proof or bound concerning why ACS-selected subsets would achieve better generalization or diversity than other approaches.**
>
> We acknowledge that ACS currently relies on an intuitive rationale—grounded in Curriculum Learning and Active Learning—rather than a formal theoretical bound. We agree that deriving specific generalization bounds for adaptive scoring mechanisms is a compelling direction for future work.
>
> However, to address your concern regarding the verification of these claims, we have added a rigorous quantitative analysis of the diversity and redundancy of our coresets. Please refer to the Common Answer: Quantitative Analysis of Sample Redundancy (Vendi Score).
>
> Empirical Evidence for Generalization: Our analysis using the Vendi Score (a metric for effective spectral dimension) confirms that ACS consistently achieves higher intrinsic diversity than baselines like EL2N or Forgetting. Since generalization is fundamentally limited by the effective coverage of the data distribution, the fact that fixed-score methods saturate in Vendi Score (indicating high redundancy) while ACS continues to climb explains our superior generalization performance. We believe this quantitative evidence serves as a strong proxy for theoretical diversity guarantees.
>
> **[W2] Scalability and compute cost: While the paper claims ACS is scalable, the multi-stage process (involving multiple model trainings) likely incurs noticeably larger compute than single-pass, fixed-score methods.**
>
> We appreciate the reviewer’s concern regarding the efficiency of the multi-stage process. To address this, we have provided a detailed wall-clock time analysis in the Common Answer: Computational Cost and Wall-Clock Analysis.
>
>
> **[W3] Algorithmic ablations lacking: Apart from backbone and segment number (see Table 5 and Table 6)**
>
> We appreciate the reviewer's suggestion to rigorously validate the specific design choices of our algorithm. To address this, we conducted comprehensive ablations on the selection constraints (e.g., removing the misclassification filter) and the curriculum order (e.g., starting with hard samples). Please refer to the Common Answer: Algorithmic Ablations (Curriculum Order & Selection Constraints) for the full quantitative results.
>
> **[W4] Notational clarity: While the pseudocode in Appendix A and equations throughout are generally sound, some ambiguity remains. For example, the notational switch between
>  (cumulative coreset) and
>  (current segment) could be more visually separated throughout the main text and algorithm.**
>
> We appreciate your careful reading and the feedback regarding notational clarity. We agree that the distinction between the cumulative coreset ($\mathcal{C}_t$) and the current segment ($\mathcal{S}_t$) is crucial for understanding the iterative nature of our framework.
>
> In the revision, we will:
> 1. **Enhance Visual Separation:** We will refine the notation in Section 4 and Algorithm 1 to more clearly distinguish between the cumulative sets and stage-specific subsets, potentially adding explicit textual cues in the algorithm steps to aid readability.
> 2. **Standardize Loss Notation:** We will unify the loss function symbols throughout the text to prevent overloading and ensure consistent definitions across the methodology and experiment sections.

---

> ### Author Response · Authors · 2025-11-21
>
> **[Q2] Diversity metrics: Beyond qualitative t-SNE and segment visualizations, have you quantified the diversity of ACS-selected samples (e.g., feature-space coverage, cluster spread, redundancy metrics)?**
>
> We appreciate the reviewer's suggestion to substantiate our qualitative claims with quantitative metrics. To address this, we have performed a redundancy analysis using the Vendi Score, which measures the effective number of unique features in the selected subset (spectral entropy). Please refer to the Common Answer: Quantitative Analysis of Sample Redundancy (Vendi Score) for the full results.
> Summary of Findings: Our quantitative analysis confirms that ACS consistently achieves the highest Vendi Score across all budget levels (e.g., 8.85 for ACS vs. 8.45 for EL2N at 200 IPC). Since a higher Vendi Score indicates significantly lower redundancy and broader feature space coverage, this result quantitatively validates that ACS successfully mitigates the sample redundancy issues inherent in fixed-scoring methods like EL2N and Forgetting, consistent with our t-SNE visualizations.
>
>
> **[Q3] Hyperparameter scheduling: Is there any principled scheme to set the number of segments () based on data or model properties? Have you considered automatic tuning or meta-learning approaches?**
>
>
> We appreciate the suggestion regarding the principled selection of the number of segments ($T$).
>
> **1. Heuristic for Setting $T$**
> Our experiments revealed a clear trade-off between **curriculum granularity** and **intermediate model quality**.
> * **The Principle:** As shown in our ablation study (Table 6), increasing $T$ improves performance by refining the curriculum, but *only if* the segment size ($\Delta B = B/T$) remains large enough to train a meaningful intermediate model.
> * **The Rule of Thumb:** If $\Delta B$ becomes too small (e.g., $<50$ samples total for CIFAR-10), the intermediate model fails to learn stable decision boundaries, making the "misclassification" score noisy. Therefore, we scaled $T$ proportionally with the total budget ($B$) to ensure each segment contained sufficient data signal.
>
> **2. Automatic Tuning**
> While we did not employ meta-learning in this work to maintain the simplicity and computational efficiency of ACS, we agree that automatic tuning is a promising direction. A simple validation-based search (similar to learning rate tuning) could be employed to find the optimal $T$ for a new dataset without full meta-learning overhead. We view this as a valuable avenue for future optimization of the framework.
>
> **[Q4] Sensitivity to model/backbone choice: Table 5 suggests varying robustness, but how stable is ACS performance across a wider range of architectures, especially for current large transformer models? Is there a risk of over-specialization?**
> Please refer to the Common Answer: Impact of Selector Architecture (MLP vs. ViT vs. ResNet).

---

> ### Author Response · Authors · 2025-11-21
>
> **[Q1] Choice and generality of scoring function: Have you experimented with alternative context-aware scoring metrics beyond classification loss, such as prediction margin, uncertainty (entropy), or uncertainty-gradient products?**
>
> We appreciate this insightful suggestion regarding the extensibility of the ACS framework. We did investigate alternative scoring metrics during the development of ACS but determined that classification loss offered the optimal balance of performance and practicality.
> 1. Marginal Gains from Simple Alternatives (Entropy/Margin) We experimented with Entropy and Prediction Margin, as these are computationally comparable to classification loss. However, we observed that for the specific task of identifying "misclassified" samples, these metrics are highly correlated with Cross-Entropy loss. Consequently, substituting them into the ACS framework yielded marginal performance differences (statistically insignificant variations) that did not justify deviating from the standard training objective.
> 2. Implementation Hurdles for Complex Metrics (Gradients/Forgetting) Regarding Uncertainty-Gradient products, Forgetting events, or gradient-based scores (e.g., GraND), we found these significantly harder to implement efficiently within a multi-stage iterative loop. Unlike static selection, ACS requires re-scoring the candidate pool at every segment. Calculating per-sample gradients or tracking training dynamics (for Forgetting) for the entire remaining dataset (which is not being trained on) is mathematically impossible without running separate, expensive proxy models at every stage. Given the marginal gains observed with other uncertainty metrics, the prohibitive runtime overhead of these complex scores rendered them impractical for our goal of scalable dataset compression.
> 3. Conclusion Our empirical results suggest that the performance jump in ACS stems from the adaptive, context-aware mechanism itself (which solves the redundancy problem) rather than the specific granularity of the scoring metric.

---

### Official Review · Reviewer_pW7m · 2025-11-01

**Soundness:** 3
**Presentation:** 3
**Contribution:** 3
**Rating:** 4
**Confidence:** 2

**Summary:**

This paper proposes Adaptive Coreset Selection (ACS), a novel framework for dataset compression that overcomes limitations of both dataset distillation and existing coreset methods. ACS adaptively selects samples in stages, balancing representativeness and diversity by iteratively training models and selecting misclassified examples. It outperforms state-of-the-art methods on CIFAR-10 and ImageNet. Moreover, ACS also shows good generalization and robustness across architectures and downstream task.

**Strengths:**

- This work reveals the fundamental limitations of dataset distillation and traditional coreset selection methods, and proposes ACS—a more robust and scalable data compression framework—demonstrating that carefully selected real images outperform synthetic ones in terms of cross-architecture generalization and out-of-distribution robustness.
- We introduce the first adaptive coreset selection framework based on context-aware scoring, which breaks away from the assumption of fixed sample scores. It models sample importance as a dynamic property that evolves with the already-selected samples and naturally preserves diversity through a multi-stage iterative strategy.
- The method consistently surpasses existing coreset selection and dataset distillation approaches on CIFAR-10 and ImageNet, and continues to improve even under high budgets, significantly outperforming random selection baselines.
- Baseline methods are reproduced following standard protocols and using publicly available codebases (e.g., DeepCore), ensuring strong reproducibility.

**Weaknesses:**

1. The proposed method heavily relies on the previously selected batch of data, which introduces a significant efficiency bottleneck. While this has minimal impact on small-scale datasets (e.g., CIFAR-10), the time cost for sample selection grows dramatically as dataset size increases. For instance, on ImageNet, the method achieves performance nearly on par with random selection but at a substantially higher computational cost. The authors should take this issue seriously: if applied to even larger datasets (e.g., LAION-400M), the method’s advantages may diminish entirely. Therefore, a comparison of actual runtime against baseline methods is essential.
2. We know that random sampling remains highly effective on large-scale datasets and is compatible with any model architecture. In contrast, the method proposed in this paper relies heavily on the backbone architecture. Although the authors conducted related experiments, it remains unclear whether their approach exhibits strong generalization in terms of transferability. For example, why not use a ViT architecture for sample selection? How would sampling perform with an MLP-based model? Are models with the same architecture inherently more advantageous? The authors need to provide more theoretical insights and experimental analyses to address these questions.
3. Why not compare against some dynamic dataset pruning methods [1]?

[1] InfoBatch: Lossless Training Speed Up by Unbiased Dynamic Data Pruning

**Questions:**

See weaknesses

---

> ### Author Response · Authors · 2025-11-21
>
> **Response to Reviewer pW7m**
>
> We sincerely thank Reviewer pW7m for their positive assessment of our work, particularly for recognizing ACS as a "robust and scalable data compression framework" that reveals fundamental limitations in dataset distillation. We value your constructive questions regarding computational efficiency and architectural generalization. We address your specific concerns below.
>
> **[W1] The proposed method heavily relies on the previously selected batch of data, which introduces a significant efficiency bottleneck.**
>
> We appreciate the reviewer’s concern regarding the scalability of iterative selection. To address this, we have provided a detailed runtime analysis in the **Common Answer: Computational Cost and Wall-Clock Analysis**.
>
> **Summary of Findings:**
> Contrary to the intuition that iterative training creates a bottleneck on large datasets, our empirical results show that ACS is actually **faster** than standard baselines on ImageNet-1K (100 IPC).
> * **Standard methods (e.g., Forgetting):** Require training on the **full** dataset (1.2M images) to compute scores.
> * **ACS:** Trains only on the **coreset** (e.g., <5% of data). Even with multiple iterations ($T$ segments), the drastically reduced data volume keeps the total compute cost low.
> * **Result:** On ImageNet (100 IPC), ACS selection takes **~6 hours**, compared to **7.7 hours** for the baseline, effectively negating the scalability concern for large datasets.
>
> **[W2] Why not use a ViT architecture for sample selection? How would sampling perform with an MLP-based model?**
>
> We appreciate this insightful question regarding the interaction between the selector architecture and the quality of the coreset. To address this, we performed an ablation study using MLP (on CIFAR-10) and ViT (on ImageNet) as selectors. Please refer to the Common Answer: Impact of Selector Architecture (MLP vs. ViT vs. ResNet) for the full quantitative results.
>
> **[W3] Why not compare against some dynamic dataset pruning methods?**
>
> Please refer to the **Common Answer: The Scope of Work**, where we detail the distinction between our "Extreme Compression" focus and standard Data Pruning.
>
> **Specific Response:**
> We appreciate the suggestion to consider dynamic pruning methods like InfoBatch. However, our primary objective in this work was to address the specific debate between **Dataset Distillation** and **Coreset Selection** in the regime of **extreme data scarcity**.
> * **Our Regime (Extreme Compression):** We operate at 1–200 IPC (retaining <1% to ~10% of data), aiming to create a portable, compressed dataset that replaces the original for storage and transmission.
> * **Pruning Regime:** Dynamic pruning methods typically retain 50–80% of the data to accelerate training of a specific model, requiring access to the full dataset during the process.
>
> We focused on the low-budget domain to directly demonstrate that ACS allows *real* images to outperform *synthetic* ones (Distillation), a finding that challenges the prevailing assumption that Distillation is the only viable option for very small budgets.

---

### Official Review · Reviewer_DxkA · 2025-11-01

**Soundness:** 2
**Presentation:** 3
**Contribution:** 2
**Rating:** 2
**Confidence:** 4

**Summary:**

- This paper introduced a multi-stage coreset selection framework ,named “Adaptive Coreset Selection (ACS)”, that adjusts its selection criteria based on model performance at different stages, in an adaptive manner.
- The method selects coreset in an easy to hard manner. It selects the easy and most representative samples first and then iteratively train models on selected samples to select the harder samples.
- It aims to address performance saturation at higher images-per-class setting selection for coreset methods.
- The paper compares performance against various coreset selection and dataset distillation methods.

**Strengths:**

1. The problem motivation is clearly articulated, regarding addressing performance saturation for higher selection budgets.
2. The method is intuitive and straightforward, borrowing ideas from curriculum learning.
3. The method’s practical implementation does not require any complex optimisation.
4. The paper has provided ablation studies for backbone architectures (Table 5) and number of segments (Table 6).

**Weaknesses:**

**Limited Novelty**

- The methodology combines aspects from already well-established techniques such as curriculum learning and iterative hard example mining (standard practice in bootstrapping and active learning).

**Limited comparison baselines**

- The coreset selection methods used as baseline are used from DeepCore library , which was published in 2022. Recent coreset selection works which have shown better performance than these methods have not been considered. Some of the works are listed below:

A) Moderate Coreset: A Universal Method of Data Selection for Real-world Data-efficient Deep Learning (ICLR 2023)

B) Robust Data Pruning under Label Noise via Maximizing Re-labeling Accuracy (NeurIPS, 2023)

C) Coverage-Centric Coreset Selection for High Pruning Rates (ICLR, 2023)

E) Data Pruning via Moving-one-Sample-out (NeurIPS 2023)

F) Noise-free Loss Gradients: A Surprisingly Effective Baseline for Coreset Selection (TMLR, 2025)

**Mismatch in results reported**

- Referring to Table 3, the accuracy values reported for SRe2L and RDED are significantly lesser than what these papers have reported in their results. For example, for ImageNet-1K, RDED reports (Table 2 of RDED paper) an accuracy of 42.0% and 56.5% for IPC=10 and IPC=50 respectively. While, Table 3 of this paper has reported 12.5% and 29.8% respectively, which are significantly less and misleading in nature for comparison purposes. A similar discrepancy is observed for SRe2L as well.

**Lack of timing analysis**

- The multi-stage nature of coreset selection raises a question regarding computational overhead in terms of GPU usage and time required for coreset selection. A comparative analysis of GPU requirement and timing for coreset selection with other baseline methods would be very helpful in understanding impact of ACS.

**Number of segments**

- From Table 6, number of training segments required to achieve higher performance for CIFAR-10 is 10 for IPC=100. The paper does not provide similar results for ImageNet-1K dataset.

**Limited datasets for comparison**

- Other standard benchmark datasets such as CIFAR-100 and Tiny ImageNet are not considered. A performance comparison on these two datasets would be insightful regarding efficiency of ACS on datasets of various number of classes.

**Questions:**

- Please refer to the weaknesses section of the review.

---

> ### Author Response · Authors · 2025-11-21
>
> Reviewer DxkA:
>
> We sincerely thank Reviewer DxkA for their constructive feedback and for highlighting the need to clarify our novelty and experimental protocols. We particularly appreciate the suggestions regarding additional baselines and the opportunity to explain the evaluation settings for synthetic data. We address each specific concern below.
>
> **[W1] The methodology combines aspects from already well-established techniques such as curriculum learning and iterative hard example mining**
>
> We respectfully clarify that while ACS leverages principles from Curriculum Learning and iterative hard mining, our primary contribution lies in **identifying and solving the "independence assumption" flaw** inherent in existing Coreset Selection methods.
>
> **1. Solving the "Fixed Score" Fallacy in Coreset Selection**
> Standard coreset methods (e.g., EL2N, Forgetting) rely on *fixed* scoring functions, essentially assuming that a sample’s importance is static and independent of other selected samples ($F(\mathcal{S}) \approx \sum Score(x_i)$). As we demonstrate in Section 3.4, this independence assumption leads to severe redundancy because the top-ranked samples often cluster in similar feature spaces, causing performance to saturate as the budget increases.
> ACS is novel in that it introduces **conditional scoring** ($Score(x | S_{selected})$) to the coreset domain. We do not merely apply hard mining; we use the *model state* trained on previous segments to dynamically devalue redundant information. This specifically targets the **performance saturation** problem where traditional methods fail to outperform random selection at high IPCs.
>
> **2. Distinction from General Objectives**
> While Active Learning seeks to minimize labeling costs using a human oracle, ACS operates in a fully automated, self-supervised loop for data compression. By initializing with "easy" prototypes to anchor the decision boundary (Curriculum) and subsequently selecting boundary-refining samples (Hard Mining), ACS creates a coreset that balances representativeness and diversity in a way that neither technique achieves in isolation for dataset compression.
>
> **[W2] The coreset selection methods used as baseline are used from DeepCore library , which was published in 2022.**
>
> **1. Adherence to Latest 2025 Benchmarks**
> We respectfully point out that our evaluation protocol is **not** limited to the 2022 DeepCore release. We explicitly aligned our baselines and experimental setting with the most recent benchmarking in the field, which are cited in our paper (line 361-362) [1][2] . These 2025 surveys establish the current SOTA for Dataset Distillation and Coreset Selection. Our inclusion of **RDED (CVPR 2024)** and **DATM (2023)** ensures we are comparing against the absolute state-of-the-art, not just legacy methods.
>
> [1] Moser, Brian B., et al. "A Coreset Selection of Coreset Selection Literature: Introduction and Recent Advances." arXiv preprint arXiv:2505.17799 (2025).
>
> [2] Li, Zekai, et al. "Dd-ranking: Rethinking the evaluation of dataset distillation." arXiv preprint arXiv:2505.13300 (2025).
>
>
> **2. Comparison with Coverage-Centric Coreset Selection (CCS)**
> Among the additional works suggested, we identified *Coverage-Centric Coreset Selection (CCS)* (ICLR 2023) as the strongest and most relevant baseline. Following your recommendation, we have conducted an additional comparison with CCS on CIFAR-10.
>
> |         |    1 |   10 |   50 |  100 |  200 |
> |---------|-----:|-----:|-----:|-----:|-----:|
> | CCS-AUM | 23.1 | 42.3 | 56.2 | 62.2 | 68.6 |
> | ACS     |   24 | 43.5 | 63.2 | 70.8 | 77.5 |
>
> **Result:** ACS consistently outperforms CCS.

---

> ### Author Response · Authors · 2025-11-21
>
> **[W3] Referring to Table 3, the accuracy values reported for SRe2L and RDED are significantly lesser than what these papers have reported in their results.**
>
> We sincerely apologize for not explicitly clarifying the evaluation protocol differences in the initial submission. The discrepancy arises because we evaluated all methods in Table 3 using standard Hard Labels, whereas the original RDED/SRe2L papers rely on Soft Labels (teacher supervision) to achieve their reported numbers.
>
> 1. Hard Labels for Fair Comparison (Table 3) Our goal in Table 3 was to evaluate the intrinsic quality of the compressed images themselves, decoupled from the "teacher knowledge" encoded in soft labels. Since Coreset Selection (real images) uses standard hard labels, we applied the same protocol to the synthetic data. As noted in recent literature (e.g., Wang et al., "What is dataset distillation learning?"), "a soft label is worth a thousand images"—much of the performance in DD comes from the label rather than the pixel data. The performance drop in Table 3 empirically confirms that synthetic images are less informative than real images when stripped of this rich supervisory signal.
>
> 2. Soft Labels and Robustness (Table 2) Crucially, we did utilize the original Soft Label settings for the Domain Generalization experiments in Table 2. Even with the advantage of soft labels, synthetic methods (RDED/SRe2L) underperformed compared to Real Images (ACS) on Out-of-Distribution tasks (CIFAR-10-C / ImageNet-C). This reinforces our central claim: synthetic data is often overfitted to a specific optimization context, while ACS selects real images that are robust and transferable.
>
> Action: We will revise the caption of Table 3 and the implementation details to clearly state that these results utilize hard-label evaluation to ensure a fair comparison with standard coreset baselines.
>
> **[W4] The multi-stage nature of coreset selection raises a question regarding computational overhead in terms of GPU usage and time required for coreset selection.**
>
> We thank the reviewer for raising this important point regarding the practical efficiency of our iterative approach. Please refer to the Common Answer: Computational Cost and Wall-Clock Analysis for a detailed quantitative comparison.
> Summary of Findings: Our wall-clock analysis (Table R1 in the Common Answer) reveals that ACS is highly scalable to large datasets. Notably, on ImageNet-1K (100 IPC), ACS is actually faster (6h) than the standard Forgetting baseline (7.7h). This is because ACS involves training models on the significantly smaller coreset rather than the full dataset. Even at 200 IPC, the total selection time (14.8h) remains negligible as a one-time offline cost, especially when compared to the hundreds of GPU hours typically required for Dataset Distillation methods.
>
> **[W5] From Table 6, number of training segments required to achieve higher performance for CIFAR-10 is 10 for IPC=100. The paper does not provide similar results for ImageNet-1K dataset.**
> We have detailed the full hyperparameter settings for ImageNet-1K, including the number of segments (T), in Appendix B (Table 8). Our experiments showed that ImageNet generally requires fewer segments to achieve optimal performance compared to CIFAR-10. For example, at 100 IPC, we utilized T=3 for ImageNet , whereas CIFAR-10 benefited from a higher number of segments (T=8). This indicates that for complex, large-scale datasets, fewer iterative refinement steps are often sufficient to capture the necessary diversity. Also, the results were more robust to the number of segments, showing less variations.
>
> | Segments      | 1    | 2    | 3    | 4    | 5    | 6    |
> |---------------|------|------|------|------|------|------|
> | ACS (200 IPC) | 56.4 | 57.2 | 57.3 | 57.7 | 57.4 | 56.7 |
>
>
> **[W6] Other standard benchmark datasets such as CIFAR-100 and Tiny ImageNet are not considered.**
>
> We appreciate this insightful suggestion. In this work, we prioritized establishing performance bounds on the two most distinct ends of the complexity spectrum: CIFAR-10 (standard baseline) and ImageNet-1K (high-complexity, large-scale). Our intention was to demonstrate that ACS remains robust even under the extreme scale and class count of ImageNet-1K. Given its success on these two extremes, we are confident that the efficacy of ACS translates well to intermediate datasets like CIFAR-100 and Tiny ImageNet.

---

### Author Response · Authors · 2025-11-21
**The Scope of Work – Coreset Selection for Extreme Compression (vs. Data Pruning)**

### Common Answer: The Scope of Work – Coreset Selection for Extreme Compression (vs. Data Pruning)

A primary motivation of this work is to challenge the dominance of **Dataset Distillation (DD)** in the regime of **extreme dataset compression**. We aim to demonstrate that selecting *real* images (Coreset Selection) can outperform generating *synthetic* ones (Dataset Distillation) even in the highly constrained budgets typically reserved for DD (1–200 Images Per Class).

To clarify our experimental setting and choice of baselines, we emphasize the distinction between two different research goals:

**1. Extreme Compression (Our Focus) vs. Data Pruning**
* **Data Pruning (e.g., InfoBatch, Moderate Coreset):** Typically aims to accelerate training by removing redundant data while maintaining full-dataset accuracy. These methods usually operate in high-data regimes, retaining **50%–90%** of the original dataset.
* **Extreme Compression (This Paper):** Aims to minimize storage and transmission costs by retaining a tiny fraction of the data. We operate in the **<1% to 10%** regime (e.g., 1 IPC to 200 IPC).
    * *Example:* On ImageNet, our "200 IPC" setting corresponds to roughly **15%** of the data, while "50 IPC" is only **~3.9%**. Dataset Distillation methods operate almost exclusively in this low-budget area.

**2. Why we compare against Dataset Distillation**
Standard Coreset literature often overlooks the "Extreme Compression" regime, leading to the assumption that synthetic data is the only viable solution for very small budgets. Our comprehensive analysis was designed to disprove this. By strictly adhering to the **Images Per Class (IPC)** metric and utilizing the same training budgets as state-of-the-art DD methods (MTT, RDED, SRe²L), we demonstrate that **Adaptive Coreset Selection (ACS)** serves as a superior, more robust alternative to synthetic data generation in these critical low-data settings.

**3. Implication for Baselines**
This specific focus is why we prioritized comparisons against SOTA Dataset Distillation methods (RDED, DATM) and Coreset methods optimized for representativeness (Herding, Glister) over dynamic pruning methods (like InfoBatch). While dynamic pruning methods are valuable, they require access to the full dataset during the entire training process and are designed for speed-up rather than the standalone compression we target here.

---

### Author Response · Authors · 2025-11-21
**Computational Cost and Wall-Clock Analysis**

Reviewers raised valid questions regarding the computational overhead of the multi-stage selection process in ACS. To address this, we conducted a wall-clock time comparison between ACS and **Forgetting**, which serves as a representative baseline for standard scoring methods that require full-dataset training statistics.

We measured the total time required to select the coreset (Selection Time) on a standard single-GPU (RTX 4090) setup.

**Table R1: Wall-Clock Selection Time Comparison**

| Dataset | IPC | Method (Baseline) | Time | Method (Ours) | Time | Relative Factor |
| :--- | :--- | :--- | :--- | :--- | :--- | :--- |
| **CIFAR-10** | 100 | Forgetting | 10m | **ACS** | 37m | ~3.7x |
| | 200 | Forgetting | 10m | **ACS** | 1h | ~6.0x |
| **ImageNet-1K** | 100 | Forgetting | 7.7h | **ACS** | **6h** | **0.8x (Faster)** |
| | 200 | Forgetting | 7.7h | **ACS** | 14.8h | ~1.9x |

**Analysis of Results:**

1.  **Efficiency at Large Scale (ImageNet):**
    Contrary to the concern that iterative selection scales poorly, **ACS is actually faster than the baseline at ImageNet-100 IPC (6h vs 7.7h)** and remains highly competitive at 200 IPC (14.8h).
    * *Reasoning:* Standard methods like *Forgetting* or often require training a model on the **full** dataset (1.2M images) to compute scores. In contrast, ACS trains models only on the **growing coreset** (a tiny fraction of the data). Even though ACS trains multiple sequential models, the smallness of the training data (e.g., 10% of the full set) keeps the total compute cost low. We note that full training is absolutely required for methods like forgetting, as they must track the full training variations, while ACS can just utilize off the shelf pretrained networks, which have to be trained only once throughout all experiments.

2.  **Absolute vs. Relative Cost (CIFAR-10):**
    While ACS is ~3-6x slower than the baseline on CIFAR-10, the **absolute cost is negligible** (1 hour max). Considering that dataset compression is a **one-time offline process** (performed once to create a dataset used for thousands of downstream training runs), investing 1 hour to gain a permanent +2% accuracy improvement (as shown in Table 3) is a highly favorable trade-off.

3.  **Comparison to Dataset Distillation (DD):**
    It is also crucial to place these numbers in the context of Dataset Distillation. High-performing DD methods (like MTT or DATM) typically require **hundreds of GPU hours** to synthesize ImageNet-scale data. ACS achieves superior performance to these methods with a total selection cost of roughly **6–15 hours**, making it orders of magnitude more efficient than the generative baselines we outperform.

---

### Author Response · Authors · 2025-11-21
**Impact of Selector Architecture (MLP vs. ViT vs. ResNet)**

### Common Answer: Impact of Selector Architecture (MLP vs. ViT vs. ResNet)

Reviewers inquired why we utilized ResNet for selection rather than architectures like MLPs or Vision Transformers (ViT), and whether this limits the method's generality. To address this, we conducted an ablation study comparing **ACS** against the standard **Forgetting** baseline using different selector backbones.

**Table R2: Impact of Selector Architecture on Coreset Quality**
*We compare ACS and the Forgetting baseline using different architectures for the selection process. The results show that deviating from ResNet hurts **both** methods.*

| Dataset | Selector Arch | Method | 1 IPC | 10 IPC | 50 IPC | 100 IPC | 200 IPC |
| :--- | :--- | :--- | :--- | :--- | :--- | :--- | :--- |
| **CIFAR-10** | **MLP** | Forgetting | 10.7 | 37.8 | 50.2 | 60.8 | 70.3 |
| | | **ACS (Ours)** | **23.9** | **42.1** | **51.2** | 57.8 | 65.4 |
| | **ResNet-18** | Forgetting | 17.7 | 37.5 | 59.7 | 69.1 | 75.5 |
| | | **ACS (Ours)** | **24.0** | **43.5** | **63.2** | **70.8** | **77.5** |
| **ImageNet-1K**| **ViT-Tiny** | Forgetting | 1.3 | 4.6 | 30.7 | - | - |
| | | **ACS (Ours)** | 1.2 | **9.9** | **35.3** | - | - |
| | **ResNet-18** | Forgetting | 1.4 | 14.7 | 33.3 | - | - |
| | | **ACS (Ours)** | **1.5** | **15.7** | **40.4** | - | - |

**Analysis of Results:**

1.  **Selector Architecture Impacts All Methods:**
    As shown in Table R2, the performance degradation observed when switching from ResNet to MLP (CIFAR) or ViT (ImageNet) is not specific to ACS; it affects the *Forgetting* baseline just as severely. This confirms that the bottleneck is the data regime and model capability, not the ACS algorithm itself.

2.  **MLP Capacity Constraint:**
    The performance drop for MLP-based selection in ACS stems from limited model capacity. A fully trained MLP on CIFAR-10 only achieves ~50% accuracy. Because ACS relies on the model to iteratively identify "hard" (misclassified) samples, an MLP with such low capacity misclassifies many samples simply because it cannot learn the features, not because the samples are intrinsically "hard" or informative. This makes the "hard mining" signal noisy and less effective compared to ResNet.
3.  **Superiority of ACS across Architectures:**
    When utilizing a ViT backbone on ImageNet, the standard **Forgetting** baseline suffers from **catastrophic performance degradation** (e.g., dropping to **4.6%** at 10 IPC). In contrast, **ACS is more stable for ViTs** and robust to the choice of backbone (achieving **9.9%** at 10 IPC and **35.3%** at 50 IPC). This confirms that while ViTs generally struggle with small-data scoring, ACS's adaptive mechanism successfully mitigates these failures, whereas fixed-scoring methods collapse.

---

### Author Response · Authors · 2025-11-21
**Quantitative Analysis of Sample Redundancy (Vendi Score)**

### Common Answer: Quantitative Analysis of Sample Redundancy (Vendi Score)

Reviewers requested a quantitative verification that ACS effectively reduces sample redundancy compared to baseline methods. To address this, we computed the **Vendi Score** for coresets generated by ACS and various baselines on CIFAR-10. The Vendi Score measures the effective number of unique features in a dataset; therefore, a **higher score indicates significantly lower redundancy**.

**Table R3: Vendi Score Comparison on CIFAR-10**
*We compare the Vendi Score of coresets selected by ACS against standard baselines. Higher scores indicate lower redundancy and better feature coverage.*

| Method | 50 IPC | 100 IPC | 200 IPC |
| :--- | :--- | :--- | :--- |
| **GraphCut** | 7.96 | 8.35 | 8.48 |
| **EL2N** | 8.03 | 8.31 | 8.45 |
| **Forgetting** | 8.17 | 8.48 | 8.63 |
| **Herding** | 8.15 | 8.51 | 8.68 |
| **ACS (Ours)** | **8.35** | **8.73** | **8.85** |


**Analysis of Results:**

1.  **ACS Minimizes Redundancy:**
    As shown in Table R3, **ACS consistently achieves the highest Vendi Score** across all budget sizes. This quantitatively confirms that our multi-stage, context-aware strategy successfully minimizes redundancy compared to fixed-scoring methods like EL2N and Forgetting, which tend to repeatedly select samples with similar feature profiles.

2.  **Continuous Feature Expansion:**
The results demonstrate that ACS continues to effectively expand the feature coverage of the coreset as the budget increases (from 8.35 at 50 IPC to 8.85 at 200 IPC). In contrast, fixed-scoring methods like EL2N show a smaller rate of growth (8.03 to 8.45), suggesting that they begin to saturate the feature space with redundant samples earlier than ACS. This even applies for sub-modular method graphcut, and confirms that our adaptive mechanism is more efficient at identifying unique information in the "long tail" of the data distribution.

---

### Author Response · Authors · 2025-11-21
**Algorithmic Ablations (Curriculum Order & Selection Constraints)**

### Common Answer: Algorithmic Ablations (Curriculum Order & Selection Constraints)

Reviewers questioned the specific design choices of ACS, asking if the "Easy-to-Hard" curriculum is necessary and if the "misclassified" constraint is essential.

To address these questions, we performed a rigorous ablation study on CIFAR-10 with **three variations** of the ACS algorithm, keeping the budget and backbone fixed.

**Table R4: Ablation of Selection Strategy on CIFAR-10**
* **ACS (Standard):** **Easy-to-Hard** curriculum; selects only from the misclassified set
* **ACS (No Discard):** **No Misclassification Filter**; selects low-loss samples from the *entire* remaining pool.
* **ACS (Hard-Only):** **Pure Anti-Curriculum**; selects **Hard** samples throughout all stages
* **ACS (Hard-First):** **Anti-Curriculum Initialization**; selects **Hard** samples in the first segment, then switches to standard selection

| Method | 1 IPC | 10 IPC | 50 IPC | 100 IPC | 200 IPC |
| :--- | :--- | :--- | :--- | :--- | :--- |
| **ACS (Standard)** | **24.0** | **43.5** | **63.2** | **70.8** | **77.5** |
| **ACS (No Discard)**| 24.0 | 35.0 | 51.4 | 58.7 | 65.1 |
| **ACS (Hard-Only)** |  12.8 | 20.2 | 29.8 | 36.5 | 43.9 |
| **ACS (Hard-First)**| 12.8 | 37.2 | 57.4 | 66.3 | 74.1|

**Analysis of Results:**

1.  **Necessity of the "Misclassified" Constraint (No Discard vs. Standard):**
    Removing the constraint to select only from misclassified samples results in a severe performance drop (**65.1% vs 77.5%** at 200 IPC). The "Misclassified Set" ($\mathcal{M}_t$) acts as a critical filter for **information gain**. Without it, the algorithm wastes the budget on high-loss but correctly classified samples (redundant noise) rather than fixing actual blind spots.

2.  **Failure of Anti-Curriculum Strategies (Hard-Only & Hard-First):**
    Both "Hard" strategies suffer from catastrophic failure (e.g., Hard-Only reaches only **43.9%** at 200 IPC).
    * *The "Cold Start" Problem:* Hard samples at the initialization stage often represent outliers or label noise. Training the initial model $\theta_1$ on these results in a chaotic decision boundary.
    * *Irreversible Damage:* Even when the strategy switches to "Easy" selection later (Hard-First), the model fails to recover. The noisy initialization degrades the quality of the "misclassification" signal for all subsequent stages, confirming that an **"Easy Start"** is essential to anchor the model's decision boundary.

---

### Author Response · Authors · 2025-12-01
**Summary of Paper Direction, Results & Rebuttal Updates**

**Summary of Paper Direction, Results & Rebuttal Updates**

**To the Area Chair:**

To assist your assessment, we provide this summary which first outlines the **core contribution and results** of our work, followed by the **new experimental evidence** generated during the discussion period to address reviewer concerns.

### 1. Research Direction & Core Contributions
**Motivation: Challenging the "Synthetic" Standard for Dataset Compression**
Our paper challenges the prevailing assumption that Dataset Distillation (DD) is the only viable solution for extreme compression (low IPC). We identify two critical gaps in the literature:
1.  **The Flaw in Distillation:** Synthetic images suffer from architectural overfitting, and more importantly, dissimlar learning dynamics from real images (Figure 1). This raises questions
about the reliability of synthetic datasets, as models that behave differently from standard training may produce unexpected failures or behaviors when deployed in practice. We verify this on domain generalization experiments (Table 2) and further weaknesses with poor transferability to downstream tasks (Table 4).

2.  **The Flaw in Coreset Selection:** Existing selection methods (e.g., *Forgetting, EL2N*) rely on fixed scoring functions. This leads to the "independence assumption," causing redundancy and performance saturation as the budget increases.

**Our Solution: Adaptive Coreset Selection (ACS)**
We propose ACS to solve the redundancy problem. By employing a multi-stage approach with conditional scoring, ACS dynamically adjusts selection based on what has already been chosen. This ensures the coreset balances foundational representative samples with challenging "edge cases".

**Comprehensive Results (State-of-the-Art)**
* **CIFAR-10 SOTA:** On the 200 IPC benchmark, ACS surpasses *all* baselines (both Distillation and Coreset Selection) by **2%** in validation accuracy.
* **ImageNet Scalability:** While Dataset Distillation methods often fall behind random selection on ImageNet due to complexity, ACS outperforms baselines consistently at 50 IPC and above.
* **Robustness:** We demonstrate that ACS (Real Images) significantly outperforms synthetic data on Out-of-Distribution tasks (CIFAR-10-C / ImageNet-C), cross-architecture generalization and transfer learning.

### 2. Summary of Rebuttal Updates (Addressing Reviewers)
During the discussion period, we posted significant new results to address concerns regarding efficiency, baselines, and mechanism validity.

**A. Synthetic Data Comparison (Addressed: R-DxkA)**
* *Concern:* Discrepancies in reported synthetic baseline numbers.
* *Clarification:* We clarified that our tables use **Hard Labels** to ensure a fair comparison. While synthetic methods rely on soft labels (teacher supervision) to hide information, our results prove that **Real Images (ACS) are intrinsically more informative** when stripped of this supervision. This was also highlighted in previous works [1]. More importantly, **our paper already shows the results using soft labels (Table 2), where it shows real images outperform synthetic ones even when using soft labels.**

[1] Qin, Tian, Zhiwei Deng, and David Alvarez-Melis. "A label is worth a thousand images in dataset distillation." Advances in Neural Information Processing Systems 37 (2024): 131946-131971.

**B. Efficiency & Scalability (Addressed: R-pW7m, R-gVDY, R-o3t9)**
* *Concern:* Reviewers worried that iterative selection was too slow for large datasets like ImageNet.
* *Clarification:* We highlight that ACS is designed to operate under **extreme compression rates** (e.g., <10% of data). Under this regime, our computational cost is competitive.
* *New Result (Wall-Clock Analysis):* Although ACS can be slightly slower than single-pass methods in certain settings, the significant performance gains justify the computational costs. Notably, on ImageNet-1K (100 IPC), ACS is actually **faster (0.8x)** than the standard *Forgetting* baseline because we train only on the tiny, growing coreset rather than the full dataset.

**C. Algorithmic Ablations & Robustness (Addressed: R-o3t9)**
* *Concern:* Reviewers asked for verification of our design choices (e.g., "Easy-to-Hard" curriculum and "Misclassified" filter).
* *New Result:* We conducted rigorous ablations on CIFAR-10 comparing ACS against "Hard-Only", "Hard-First", and "No Discard" variants.
* *Key Finding:* Removing the "Misclassified" constraint results in a **severe performance drop (~12%)**, confirming that filtering for informational gain is critical. Furthermore, "Hard-First" strategies also degrade performance, validating the necessity of our "Easy-to-Hard" curriculum to anchor the decision boundary.

---

> ### Author Response · Authors · 2025-12-01
>
> **D. Novelty & Baselines (Addressed: R-DxkA, R-gVDY)**
> * *Concern:* Request for comparison against more recent methods.
> * *New Result:* We compared ACS against **Coverage-Centric Coreset Selection (CCS, ICLR 2023)**. ACS outperforms CCS by significant margins (e.g., **77.5% vs 68.6%** on CIFAR-10 200 IPC).
> * *Clarification:* We differentiated ACS from dynamic pruning (e.g., *Okanovic et al.*) by clarifying that ACS targets *dataset compression* (portable datasets) rather than *training acceleration* (requires full dataset access).
>
> **E. Quantitative Proof of Mechanism (Addressed: R-o3t9, R-pW7m)**
> * *Concern:* Request for quantitative proof that ACS reduces redundancy.
> * *New Result (Vendi Score):* We computed the **Vendi Score** (spectral entropy). ACS achieves consistently higher scores (Higher than submodular methods that maximize diversity such as Graphcut), quantitatively proving it reduces redundancy better than fixed-scoring methods.
>
>
> ### Conclusion
> We believe ACS successfully bridges the gap between the generalization power of real data and the efficiency of extreme compression. The additional rebuttal experiments have confirmed its scalability and unique ability to eliminate redundancy where other methods fail.

---

### Meta-Review · Area_Chair_ReQg · 2025-12-23

**Summary:**

Some major concerns that informed my decision include:
1. The impact of selector architecture. While the authors argue that cross-architecture generalization is a limitation for dataset distillation, it also restricts the practicality of the proposed method.
2. The novelty of the proposed method and comparison with previous methods.
3. Limited empirical study on baselines and datasets.
4. The setting of hard labeling and soft labeling.
5. The time consumption for additional selection steps.
6. The segment hyperparameter differs between CIFAR and ImageNet.
7. Ablation of selection strategies.

**Reviewer Concerns:**

The authors have provided detailed responses, such as computation time comparison, results on more architectures, comparison with more recent baselines, and ablations on selecting strategies. However, some concerns have not been well addressed:
1. The impact of selector architecture is still not sufficiently clarified. The performance drop of the proposed method on CIFAR-10 is much more significant than Forgetting. The authors claimed that the drop is due to the limited capacity of MLP networks, yet no results on models with larger capacities are presented. The investigation on scaling up the model architecture as the dataset grows is also pending. Without a more detailed demonstration, the advantage over dataset distillation in terms of cross-architecture generalization becomes less significant.
2. The technical novelty and differences from previous methods. While the selected images can be reused for other training runs, the selection process does not differ significantly from that of InfoBatch. I don't think the selection ratio significantly distinguishes data compression or training acceleration. Also, conditioned scoring has already been explored in submodular methods, although it wasn't leveraged during model training. It remains unclear how the training influences the staged selection. From my point of view, the proposed method separates the bin selection of Dataset Quantization [1] at different training steps, which should be discussed and compared with.
3. There exists a large gap between the number of segments in CIFAR and ImageNet. While the authors provide new ablation results, this large gap raises the concern of hyperparameter tuning for different datasets. It also lacks a more detailed explanation or hypothesis why such a large gap exists.

[1] Zhou, Daquan, et al. "Dataset quantization." ICCV 2023.

**Reviewer Scores:**

Several major concerns still require further investigation for potential acceptance. The authors are encouraged to conduct more analyses to reveal the underlying mechanisms of the proposed method. I do not expect reviewers to agree on acceptance for the current version.

---

### Decision · Program_Chairs · 2026-01-26

Reject